# FetCAT: Cross-attention fusion of transformer-CNN architecture for fetal brain plane classification with explainability using motion-degraded MRI

**Sayma Alam Suha**[iD]*, **Rifat Shahriyar**[iD]

Department of Computer Science and Engineering, Bangladesh University of Engineering and Technology, Dhaka, Bangladesh

* suha@bup.edu.bd

## Abstract

Fetal brain magnetic resonance imaging (MRI) has been recognized as a vital diagnostic tool for identifying neurological anomalies during pregnancy. Accurate classification of fetal MRI planes is essential for effective prenatal neurological assessment, yet this task remains challenging in clinical practice. Key obstacles include the reliance on manual identification by specialized neuroradiologists, resource-constraints, motion-induced artifacts from fetal movement, and insufficient clinical interpretability of automated methods. This study presents FetCAT (Fetal Cross-Attention Transformer), a novel hybrid architecture that integrates a pre-trained Swin Transformer with a custom AdaptiveMed-CNN model through cross-attention fusion mechanisms for automated fetal brain MRI plane classification. The proposed hybrid architecture combines the global contextual understanding capabilities of transformers with the local feature extraction strengths of CNN through a sophisticated cross-attention mechanism. The model was trained and tested with a large-scale dataset of 52,561 motion-degraded fetal MRI slices from 741 patients, encompassing three anatomical planes and a gestational age of 19-39 weeks. Comprehensive comparative analyses were conducted across pre-trained CNN architectures, baseline and pre-trained transformer models, and the proposed hybrid configurations to evaluate the efficacy. Systematic ablation studies were performed to evaluate the impact of domain-specific data augmentation strategies on model performance. Robust statistical evaluation, including mean, variance, confidence intervals, and McNemar's test, substantiated the significant performance advantage of the proposed architecture over all competing models. Additionally, Grad-CAM-based explainability analysis was implemented to provide visual interpretations of the model's decision-making process, thereby enhancing clinical interpretability. The proposed cross-attention based

**Data availability statement:** The fetal brain MRI dataset analyzed for train and test in this study is publicly available from the Stanford University Digital Repository at https://purl.stanford.edu/sf714wg0636 [35], with no access restrictions, users can download the anonymized .jpg images directly upon visiting the link. The MRI dataset used for test set is collected from OpenNeuro fetal MRI publicly available repository at fetal-fMRI-OpenNeuro (https://openneuro.org/datasets/ds003090/versions/1.0.0/file-display/task-rest_bold.json). The source code for the hybrid Swin Transformer-CNN architecture and implementation used in this study is publicly available at: https://github.com/SuhaAlam/FetCAT.

**Funding:** The author(s) received no specific funding for this work.

**Competing interests:** The authors have declared that no competing interests exist.

Swin-AdaptiveMedCNN model achieved superior performance with 98.64% accuracy without data augmentation, substantially outperforming standalone CNN models, baseline and pre-trained transformers. Explainability analysis using Grad-CAM visualization demonstrated that the model focuses on clinically relevant anatomical landmarks. Contrary to common assumptions, ablation studies revealed that data augmentation consistently reduced model performance rather than improving it. This result can be attributed to the inherent diversity and natural variability already present in the dataset, which rendered additional synthetic variations counterproductive. Moreover, the proposed FetCAT model also demonstrated strong generalization capability, maintaining superior and statistically significant performance on an unseen OpenNeuro MRI test dataset with 81.0% accuracy. Thus, this study establishes a benchmark for automated fetal brain MRI plane classification.

## 1 Introduction

Fetal brain analysis represents a fundamental aspect of maternal healthcare, providing crucial information about neurological development during pregnancy [1]. Rapid neurodevelopmental changes that occur in the fetus require early detection of abnormalities to allow prompt medical intervention. Magnetic Resonance Imaging (MRI) serves as an essential diagnostic tool for identifying neurological anomalies and developmental disorders affecting fetal brain growth [2]. Recent advances in fetal MRI have revealed that prenatal exposures can significantly disrupt brain development and increase the risk of neuropsychiatric disorders [3]. In such cases, the precise classification of the Axial, Coronal, and Sagittal fetal brain magnetic resonance planes is fundamental to accurate diagnosis for other downstream tasks [4]. Each imaging plane provides unique anatomical perspectives: Axial planes display horizontal cross-sections vital for ventricular assessment, Coronal planes reveal anterior-posterior structures essential for corpus callosum evaluation, and Sagittal planes offer lateral profiles critical for cerebellar and brainstem examination [5]. Thus, accurate plane identification is crucial because it enables reliable automation for downstream fetal imaging tasks and supports early clinical decision-making [6]. In general, fetal brain planes in MRI are manually identified by radiologists using anatomical landmarks. However, MRI's contain superior soft tissue contrast and volumetric capabilities. Traditional fetal brain MRI analysis relies heavily on specialized neuroradiologists for accurate plane identification and interpretation [7]. This expertise is scarce, especially in remote and resource-limited regions with limited access to trained professionals [8]. Additionally, motion-related challenges in MRI are critical to address which may results into diagnostic delays in prenatal conditions analysis [9]. In such circumstances, advanced AI-based techniques for automated classification of fetal MRI planes can standardize interpretation, reduce analysis time, and extend specialized care to underserved regions [10].

However, despite recently conducted research on fetal MRI data for automated anatomical structure segmentation [11], motion correction [12], diagnosis of abnormalities and related tasks [13]; research on plane classification using motion-degraded datasets is scarce. Moreover, explainable AI (XAI) approaches remain underexplored in this domain which is essential for clinical trust and thus limiting interpretability and applicability in automated systems. A further overlooked aspect is the impact of data augmentation, which while widely adopted to enhance generalization in deep learning, shows inconsistent outcomes across medical imaging studies. While augmentation has improved performance in tasks like Polycystic Ovary Syndrome detection [14] and Alzheimer's classification [15], contradictory findings indicate negligible or negative effects in medical imaging, such as brain tumor detection [16] and COVID-19 classification [17] etc. Moreover, recent advances in vision transformers, with their strong ability to capture long-range spatial relationships, have shown superior performance over traditional CNNs, which is rarely explored in this domain [18]. Thus, fetal brain MRI plane classification, especially in motion-degraded datasets constitutes an emerging field of investigation. Given these contradictions and the unique challenges of fetal neuroimaging, this study addresses these gaps by proposing and evaluating classification performance with explainability using state-of-the-art transformer based hybrid techniques. Also, the study conducts a systematic ablation analysis of augmentation strategies to develop more reliable and transparent methods for fetal brain MRI plane recognition.

Therefore, the objective of this study is to develop and evaluate an explainable AI framework for automated classification of fetal brain MRI planes (Axial, Coronal, Sagittal) in motion-degraded datasets. To attain this objective, a novel hybrid architecture termed FetCAT has been developed, integrating a pre-trained Swin Transformer [19] with a proposed AdaptiveMed-CNN architecture through cross-attention fusion mechanisms. A cross-attention fusion mechanism was incorporated to integrate the local spatial textures captured by CNNs with the global contextual embeddings expected by transformers, thereby enabling a robust predictive model. The study hypothesizes that a cross-attention fusion mechanism, which integrates the global contextual understanding of transformers with the local feature extraction of CNNs, will achieve superior classification performance while maintaining clinical interpretability. Comprehensive comparative analysis has been conducted across pre-trained CNN architectures, baseline and pre-trained transformer models (Vision Transformer (ViT) [20], Bidirectional Encoder representation from Image Transformers (BEiT) [21], Data-efficient image Transformers (DEiT) [22], Swin transformer), and various hybrid configurations combining different transformer backbones with CNN models through the proposed fusion framework. Systematic ablation studies have been performed to evaluate data augmentation impact, while Grad-CAM-based explainability analysis has been implemented to enhance clinical interpretability. To assess the reliability, reproducibility, and calibration of the proposed FetCAT model's performance, comprehensive statistical analyses were conducted across three independent runs with 2-fold cross-validation. These analyses included 95% confidence intervals, coefficients of variation, class-wise metrics, and evaluation of Expected Calibration Error and Brier Score. Moreover, to evaluate the generalization capability of the proposed FetCAT model, a rigorous performance assessment was conducted on an unseen OpenNeuro MRI test dataset, comparing it against baseline models and confirming statistical significance via McNemar's test. Thus, the key contributions of this study are threefold:

- Firstly, FetCAT (Fetal Cross-Attention Transformer), a hybrid architecture that integrates a pre-trained Swin Transformer with a custom AdaptiveMed-CNN through cross-attention fusion mechanisms, is proposed and specifically designed for motion-degraded fetal brain MRI plane classification. To evaluate the efficacy of this model, various architectural variations are systematically tested and benchmarked.
- Secondly, comprehensive explainability analysis using Grad-CAM visualization is conducted to enhance clinical interpretability and trust in automated diagnostic systems.
- Thirdly, systematic ablation studies evaluating the impact of domain-specific data augmentation strategies on classification performance are performed, providing evidence-based guidelines for preprocessing motion-degraded fetal MRI datasets.

The remainder of this paper is structured as follows: Sect 2 reviews related works, Sect 3 presents the methodology including the proposed FetCAT architecture and experimental setup, Sect 4 discusses results and performance analysis, and Sect 5 concludes with discussion and future directions.

## 2 Background study

Fetal imaging represents a cornerstone of contemporary prenatal care, with ultrasonography (USG) and magnetic resonance imaging (MRI) constituting the primary diagnostic modalities for comprehensive fetal assessment and anomaly detection. While ultrasonography remains the gold standard for routine prenatal screening due to its accessibility and cost-effectiveness, fetal MRI has emerged as an increasingly indispensable imaging technique that offers superior soft tissue contrast, multiplanar imaging capabilities, and enhanced visualization of complex anatomical structures [31]. Thus, MRI has gained particular prominence for its ability to identify fine neuroanatomical features, structural abnormalities, and developmental differences that may be difficult to detect using conventional ultrasound methods [32]. However, the interpretation of fetal brain MRI requires substantial clinical expertise and specialized training among radiologists, making accurate analysis challenging in routine clinical practice [33].

In recent years, the exceptional diagnostic potential of fetal MRI has attracted considerable research attention. Several researchers have focused on developing advanced computational methods and machine learning approaches for automated analysis of fetal MRI images. The recent works in this domain are summarized in Table 1. This table provides an overview of 12 automated fetal brain MRI analysis studies, demonstrating diverse research objectives including brain extraction and localization, anatomical segmentation and biometry, quality assessment, and reconstruction from fetal MRI dataset. The studies collectively span various methodological approaches from traditional machine learning techniques to deep learning architectures, addressing fundamental preprocessing and analysis tasks essential for automated fetal brain imaging workflows. However, the reviewed studies on automated fetal brain MRI analysis face several challenges. Such as, small dataset sizes, restrict generalizability, particularly for rare or abnormal cases (e.g., Gopikrishna et al. [11]; Ebner et al. [29]). Many methods are constrained to specific imaging protocols (e.g., SSFSE in Ebner et al. [27]; Coronal T2WI in She et al. [13]) which limits applicability across diverse datasets. Challenges with severe motion artifacts or pathologies often lead to segmentation or reconstruction failures (e.g., Chen et al. [34]; Ebner et al. [29]). Additionally, some studies note computational constraints, such as slow reconstruction times or GPU memory limits.

Based on the existing literature, several critical research gaps are identified that necessitate targeted investigation. First, while studies have extensively focused on segmentation [13,27] and reconstruction [29,30], a notable absence is observed in the development of comprehensive *plane classification* methodologies. This foundational task is crucial for structuring volumetric analysis and facilitating motion correction in fetal MRI, yet it remains underrepresented. Second, the field lacks adequate exploration of *explainability and interpretability* mechanisms. Current approaches predominantly focus on performance metrics [11,26] without providing clinicians with meaningful insights into the automated decision-making processes, thereby limiting clinical trust and adoption. Third, there is an insufficient systematic investigation into *data augmentation strategies* specifically tailored for fetal brain morphology. These techniques have not been rigorously evaluated for their impact on generalizability across key fetal imaging variables, and their utility remains a subject of debate in the wider medical imaging field due to inconsistent outcomes [16,17].

To directly address these gaps, the proposed FetCAT framework is proposed in this work. The absence of a dedicated plane classification system is countered by the introduction of a novel hybrid Transformer-CNN architecture, which is specifically engineered for this task using a cross-attention fusion mechanism. The critical need for clinical interpretability is met through the integration of Gradient-weighted Class Activation Mapping (Grad-CAM), providing visual explanations for model predictions. Finally, the uncertainty regarding data augmentation is systematically investigated via rigorous ablation studies, thereby establishing evidence-based guidelines for preprocessing motion-degraded fetal MRI datasets.

**Table 1**. Overview of automated fetal brain MRI analysis studies.

| Author(s) (Year) | Study Objective | Used Dataset | Methodology | Key Findings |
|---|---|---|---|---|
| Li et al. (2025) [12] | Develop FetDTIAlign for dMRI registration. | 77 BCH, 40 dHCP subjects (22–36 weeks); DTI atlases. | Dual-encoder; affine/deformable registration; validated on 60 tracts. | Superior tract/FA alignment vs. FSL, DTI-TK, VoxelMorph; generalizable. |
| Gopikrishna et al. (2024) [11] | Classify ventriculomegaly; compare U-Net, DeeplabV3+. | 15 T1W Open Neuro MRIs, expert-segmented. | U-Net, DeeplabV3+ segmentation; contour-based size estimation; >10 mm threshold. | DeeplabV3+ Dice >0.8; U-Net accuracy >0.98; effective classification. |
| Vahedifard et al. (2023) [23] | Automate ventriculomegaly detection via 2D-3D measurements. | FeTA 2022 (80 T2W); 22 institutional T2W MRIs. | UNet segmentation; 3D reconstruction; ventricle measurement; >10 mm classification. | 95% accuracy; <1.7 mm error; AI matches neuroradiologist. |
| She et al. (2023) [13] | Automate 2D brain biometry using DL. | 268 T2WI SSFSE, GA 21–38 weeks. | nnU-Net for cerebrum, cerebellum, ventricles; calculate CBPD, TCD, LAD/RAD. | High correlations (0.719–0.990); low mean errors (−2.405 to −0.008 mm). |
| Shen et al. (2022) [24] | Predict Gestational Age using attention-guided DL. | 741 T2W MRIs (19–39 weeks); 156–189 external scans. | ResNet-50 CNN with attention; fine-tuned for external data. | $R^2$ 0.945, MAE 6.7 days; external $R^2$ 0.81–0.90. |
| Chen et al. (2021) [25] | Automate 3D brain extraction via DL framework. | 77 T2W MRIs (85 total, 8 excluded). | Densely-connected U-Net; local refinement; fusion network. | Superior DC, HD95 vs. U-Net, DU-Net; artifact-robust. |
| Largent et al. (2021) [26] | Assess 3D MRI quality using multi-instance DLMs. | 271 exams, GA 30.9±5.5 weeks, T2W SSFSE. | MI-CB/VB/FE-DLM; expert labels; GA input option. | MI-CB-DLM: 0.85 acc, AUC 0.93; GA improves to 0.86. |
| Ebner et al. (2020) [27] | Automate localization, segmentation, SRR, visualization. | 268 MRIs: normal, spina bifida; two scanners. | Loc-Net, Seg-Net; outlier-robust SRR; template alignment. | Fast (2.35s); high DSC; manual-quality recon; pathology-robust. |
| Li et al. (2020) [28] | Automate 2D brain extraction using DL. | 88 MRIs, GA 20–30 weeks; SSFSE; 10 cross-dataset. | Shallow FCN for ROI; deep M-FCN (110 layers) segmentation. | 100% localization; Dice 0.958; outperforms U-Net, others; 6 s/stack. |
| Ebner et al. (2018) [29] | Automate localization, segmentation, reconstruction. | 16 MRIs with spina bifida, ventriculomegaly. | P-Net for localization; multi-scale CNN; outlier-robust SRR. | High IoU, DSC 93.87%; clear SRR; outperforms SOA. |
| Tourbier et al. (2017) [30] | Automate template-based brain localization, extraction. | 66 MRIs (40 healthy, 26 pathological), GA 22–38 weeks. | Template-to-slice matching; B-spline registration; SR reconstruction. | Dice 94.5%; better PSNR vs. rigid; manual-quality recon. |

This approach is designed to provide a more comprehensive and clinically translatable solution for automated fetal brain MRI analysis.

## 3 Methodology

The methodology for the classification of the fetal brain MRI plane applied in this study is shown in Fig 1. This framework encompasses data collection, preprocessing, model development, explainable AI integration and evaluation, with detailed descriptions provided in the following subsections.

### 3.1 Data collection and preprocessing

**Train and validation set:** The fetal MRI images employed in this investigation were sourced from the publicly accessible dataset maintained by Stanford University Digital Repository collected from Stanford Lucile Packard Children's Hospital [35] (https://purl.stanford.edu/sf714wg0636). The dataset was accessed on 30 January 2025 from the Stanford University Digital Repository and was fully anonymized, with no access to personally identifiable information at any stage.

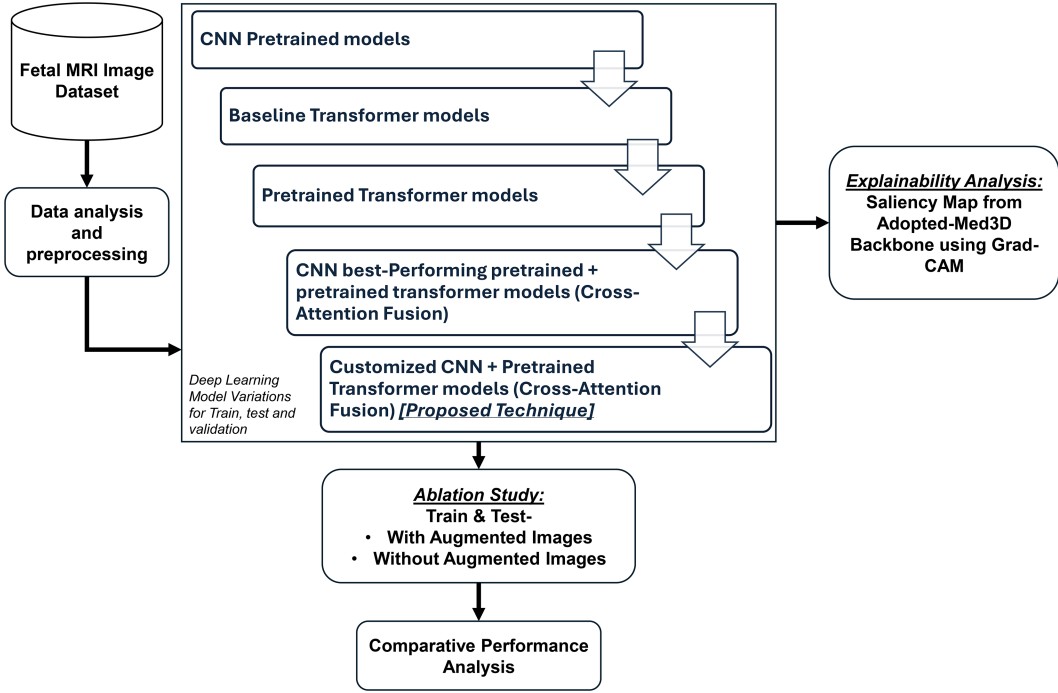

**Fig 1**. Methodological framework.

This extensive dataset encompasses a collection of clinically relevant fetal brain MRI scans that were acquired during routine medical examinations, incorporating data from 741 patients. These scans were accompanied by corresponding gestational ages, spanning from 19 to 39 weeks, which were determined based on estimated delivery dates derived from first-trimester ultrasound measurements. Each patient case was comprised of multiple imaging planes including Axial, Coronal, and Sagittal views, thereby providing comprehensive representation of fetal brain anatomy from various perspectives. The repository contained anonymized fetal MRI data that had been collected with appropriate ethical considerations and research permissions, rendering it suitable for algorithm development and validation purposes. Importantly, this dataset comprises exclusively developmentally normal fetal brain T2-weighted MRIs acquired during routine prenatal examinations, as confirmed by the original collection protocol and used in other studies [24]. No cases with confirmed neurological anomalies were included, ensuring a focus on standard anatomical variability across gestational ages without confounding pathological features. Selection criteria prioritized diagnostic-quality scans from uncomplicated pregnancies, with exclusions applied solely for technical quality issues (e.g., severe motion artifacts, low signal-to-noise ratio, or incomplete coverage) rather than clinical pathology. This composition aligns with the study's emphasis on automated plane classification as a foundational step for broader fetal neuroimaging workflows, including potential downstream anomaly detection. Consequently, the deep learning models were trained and validated in this study solely on normal scans, achieving robust performance for plane identification in this context. Evaluation on anomalous cases was beyond the current scope due to the dataset's design. Future extensions of this study could incorporate anomaly-specific datasets to assess generalization.

The acquired images were subjected to systematic analysis beginning with quality assessment to ensure diagnostic adequacy. Images exhibiting severe motion artifacts, insufficient signal-to-noise ratio, or incomplete anatomical coverage were excluded from further processing to preserve dataset integrity. The remaining images were categorized according to their anatomical planes (Axial, Coronal, and Sagittal) based on visible anatomical landmarks. The plane-wise labeling was

provided by the original dataset source. Overall, the dataset comprised a total of 52,561 fetal MRI images, which were distributed across three anatomical planes: 16,881 Axial view images, 16,534 Sagittal view images, and 19,146 Coronal view images. To ensure robust model evaluation, the fetal MRI dataset was organized into separate training and testing directories, with each directory containing three subdirectories corresponding to the anatomical planes (Axial, Coronal, Sagittal). This organizational structure ensures proper train-test separation and prevents data leakage during model development and evaluation.

The fetal brain MRI dataset analyzed for train and test in this study is publicly available from the Stanford University Digital Repository at https://purl.stanford.edu/sf714wg0636 [35], with no access restrictions, users can download the anonymized .jpg images (52,561 slices from 741 patient cases, 19–39 weeks GA) directly upon visiting the link. The MRI dataset used for test set is collected from OpenNeuro fetal MRI publicly available repository at fetal-fMRI-OpenNeuro. The source code for the hybrid Swin Transformer-CNN architecture and implementation used in this study is publicly available at: https://github.com/SuhaAlam/FetCAT.

**Test Set:** To further validate the generalizability and robustness of the proposed FetCAT model, an external test was conducted using an independent, publicly available fetal MRI dataset from OpenNeuro (accession number: ds003090) [36]. This dataset comprises resting-state BOLD fMRI scans, presenting a distinct challenge compared to the primary T2-weighted structural MRI training data. The use of BOLD fMRI data tests the model's ability to classify anatomical planes under different contrast mechanisms and potential noise profiles. From the 173 available subjects in this repository, a single midslice was systematically extracted for each of the three fundamental anatomical planes (Axial, Coronal, and Sagittal). This process yielded a total of 519 meticulously annotated images for external validation. This test set effectively simulates a real-world scenario where a model encounters data from a different institution and acquisition protocol.

**3.1.1 Image preprocessing.** The fetal MRI dataset was subjected to rigorous preprocessing to standardize the input for the neural network and ensure optimal feature extraction. Initially, all images were resized to uniform dimensions of 224×224 pixels to maintain consistency across the dataset and accommodate the input requirements of the deep learning model architecture. Subsequently, image normalization was applied using ImageNet mean values (0.485, 0.456, 0.406) and standard deviation values (0.229, 0.224, 0.225), which standardized pixel intensities and facilitated convergence during model training. This normalization step was essential for knowledge transfer from pre-trained models, as it aligned the distribution of fetal MRI images with that of the original training data. The dataset was then strategically partitioned using a stratified split approach, whereby 80% was allocated for training and 20% for validation. This stratification ensured proportional representation of each fetal brain plane category in both subsets, thereby mitigating potential bias in model training and evaluation. The validation set was carefully isolated from the training process to provide an unbiased assessment of the study. The dataset consisted of raw, non-reconstructed 2D fetal MRI slices that retained motion-induced artifacts, reflecting real-world clinical imaging conditions. No super-resolution or motion correction techniques were applied during preprocessing.

To ensure a representative distribution of anatomical planes across the training and validation splits, a stratified 2-fold cross-validation approach was employed. The validation set in Fold 1 contained 8,372 Axial, 9,177 Coronal, and 8,014 Sagittal images, while Fold 2's validation set contained 8,440 Axial, 9,573 Coronal, and 8,267 Sagittal images. This close alignment with the overall dataset distribution (Axial: 16,881, Coronal: 19,146, Sagittal: 16,534) confirms that the class ratios were preserved in both splits, preventing bias and ensuring robust evaluation of model performance. The data was partitioned using a subject-level split across the 741 unique subjects for the 2-fold cross-validation, which guaranteed that all image slices from any single patient were exclusively contained within one fold (either training or validation) to effectively prevent data leakage.

## 3.2 Proposed model: FetCAT – transformer-CNN cross-attention framework

To ensure reproducibility and provide a clear technical foundation for the hybrid approach, the detailed algorithmic steps and equations underlying the FetCAT framework are presented (Algorithm 1 and 2). Conceptually, a Swin Transformer and a custom AdaptiveMed-CNN are integrated in FetCAT via a cross-attention mechanism. Global contextual relationships are captured by the transformer, while local anatomical features are extracted by the CNN. These features are fused using a cross-attention module that enables focus on clinically relevant regions, thereby combining the strengths of both architectures. The overall system architecture of the proposed model is shown in Fig 2.

**Algorithm 1 FetCAT: Integrated transformer-CNN cross-attention for fetal brain MRI classification.**

1: **Input:** Dataset $\mathcal{D} = \{(I_i, y_i)\}_{i=1}^{N}$, $I_i \in \mathbb{R}^{H \times W \times 3}$, $y_i \in \{1, \ldots, C\}$, folds $K$, epochs $E$, batch size $B$, learning rate $\eta$
2: **Output:** Model $\Theta^*$, metrics $M = (\mu_{\mathrm{acc}}, \sigma_{\mathrm{acc}}, \mu_{\mathrm{loss}})$
3: **Initialization**
4: Initialize transformer $\mathcal{T}_{\mathrm{pre}}$, dim $d_{\mathrm{trans}}$, frozen embeddings
5: Initialize AdaptiveMedCNN architecture $\mathcal{F}_{\mathrm{med}}$: Conv-BN-ReLU-MaxPool layers, dim $d_{\mathrm{cnn}} = 512$
6: Initialize cross-attention $\mathcal{A}_{\mathrm{cross}}(Q, K, V) = \mathrm{softmax}\left(\frac{QK^T}{\sqrt{d_{\mathrm{trans}}}}\right)V$, $h = 8$
7: Initialize projection $\mathcal{P}_{\mathrm{cnn}}$, fusion $\mathcal{F}_{\mathrm{fusion}}$, classifier $\mathcal{C}$
8: **K-Fold Training**
9: **for** $k \in \{1, \ldots, K\}$ **do**
10:     $(\mathcal{D}_{\mathrm{train}}^{(k)}, \mathcal{D}_{\mathrm{val}}^{(k)}) \leftarrow \mathrm{KFoldSplit}(\mathcal{D}, k)$
11:     Initialize $\Theta^{(k)}$, AdamW($\eta = 10^{-4}$), CosineAnnealingLR
12:     **for** $e \in \{1, \ldots, E\}$ **do**
13:         **for** batch $\mathcal{B} \subset \mathcal{D}_{\mathrm{train}}^{(k)}$ **do**
14:             Preprocess: $I^{\mathrm{trans}}$, $I^{\mathrm{cnn}}$
15:             Features: $F^{\mathrm{trans}} \leftarrow \mathcal{T}_{\mathrm{pre}}(I^{\mathrm{trans}})$, $F^{\mathrm{cnn}} \leftarrow \mathrm{GlobalAvgPool}(\mathcal{F}_{\mathrm{med}}(I^{\mathrm{cnn}}))$
16:             Project: $F^{\mathrm{cnn}'} \leftarrow \mathcal{P}_{\mathrm{cnn}}(F^{\mathrm{cnn}})$
17:             cross-attention: $A_{\mathrm{cross}} \leftarrow \mathcal{A}_{\mathrm{cross}}(F^{\mathrm{trans}}, F^{\mathrm{cnn}'}, F^{\mathrm{cnn}'})$
18:             Fuse: $F_{\mathrm{fused}} \leftarrow \mathcal{F}_{\mathrm{fusion}}(F^{\mathrm{trans}} \oplus A_{\mathrm{cross}})$
19:             Logits: $\hat{y} \leftarrow \mathcal{C}(F_{\mathrm{fused}})$
20:             Loss: $\mathcal{L}_{\mathrm{CE}} \leftarrow -\frac{1}{B} \sum_{i=1}^{B} \sum_{c=1}^{C} y_{i,c} \log \hat{p}_{i,c}$
21:             Update $\Theta^{(k)}$ with clipped gradients
22:         **end for**
23:         Evaluate $\mathcal{L}_{\mathrm{val}}, A_{\mathrm{val}}$ on $\mathcal{D}_{\mathrm{val}}^{(k)}$
24:         **if** $A_{\mathrm{val}} > A_{\mathrm{best}}^{(k)}$ **then**
25:             Save $\Theta^{(k)*}$, reset patience
26:         **end if**
27:     **end for**
28:     Store $A_{\mathrm{best}}^{(k)}, \mathcal{L}_{\mathrm{best}}^{(k)}$
29: **end for**
30: **Selection and Testing**
31: $\Theta^* \leftarrow \Theta^{(\arg\max_k A_{\mathrm{best}}^{(k)})*}$
32: Compute $M = (\mu_{\mathrm{acc}}, \sigma_{\mathrm{acc}}, \mu_{\mathrm{loss}})$ across folds
33: Evaluate $\Theta^*$ on $\mathcal{D}_{\mathrm{test}}$ for test metrics
34: **Inference**
35: **function** Predict($I_{\mathrm{new}}, \Theta^*$)
36:     $F^{\mathrm{trans}}, F^{\mathrm{cnn}} \leftarrow \mathcal{T}_{\mathrm{pre}}, \mathcal{F}_{\mathrm{med}}$
37:     $F_{\mathrm{fused}} \leftarrow \mathcal{F}_{\mathrm{fusion}}(\mathcal{A}_{\mathrm{cross}}(F^{\mathrm{trans}}, \mathcal{P}_{\mathrm{cnn}}(F^{\mathrm{cnn}})))$
38:     **return** $\arg\max(\mathrm{softmax}(\mathcal{C}(F_{\mathrm{fused}})))$
39: **end function**
40: **return** $\Theta^*, M$

## Algorithm 2 Feature fusion mechanism.

```
1: procedure FUSEFEATURES(f_t, f_c)
2:     Project CNN features:  f'_c ← W_c f_c + b_c
3:     Compute attention:
4:         Q ← W_q f_t
5:         K ← W_k f'_c
6:         V ← W_v f'_c
7:         A ← Softmax( QK^T / √d )
8:         f_a ← A · V
9:     Concatenate:  f_cat ← [f_t ‖ f_a]
10:     Fuse:  f_f ← ReLU(W_f f_cat + b_f)
11:     return f_f
12: end procedure
```

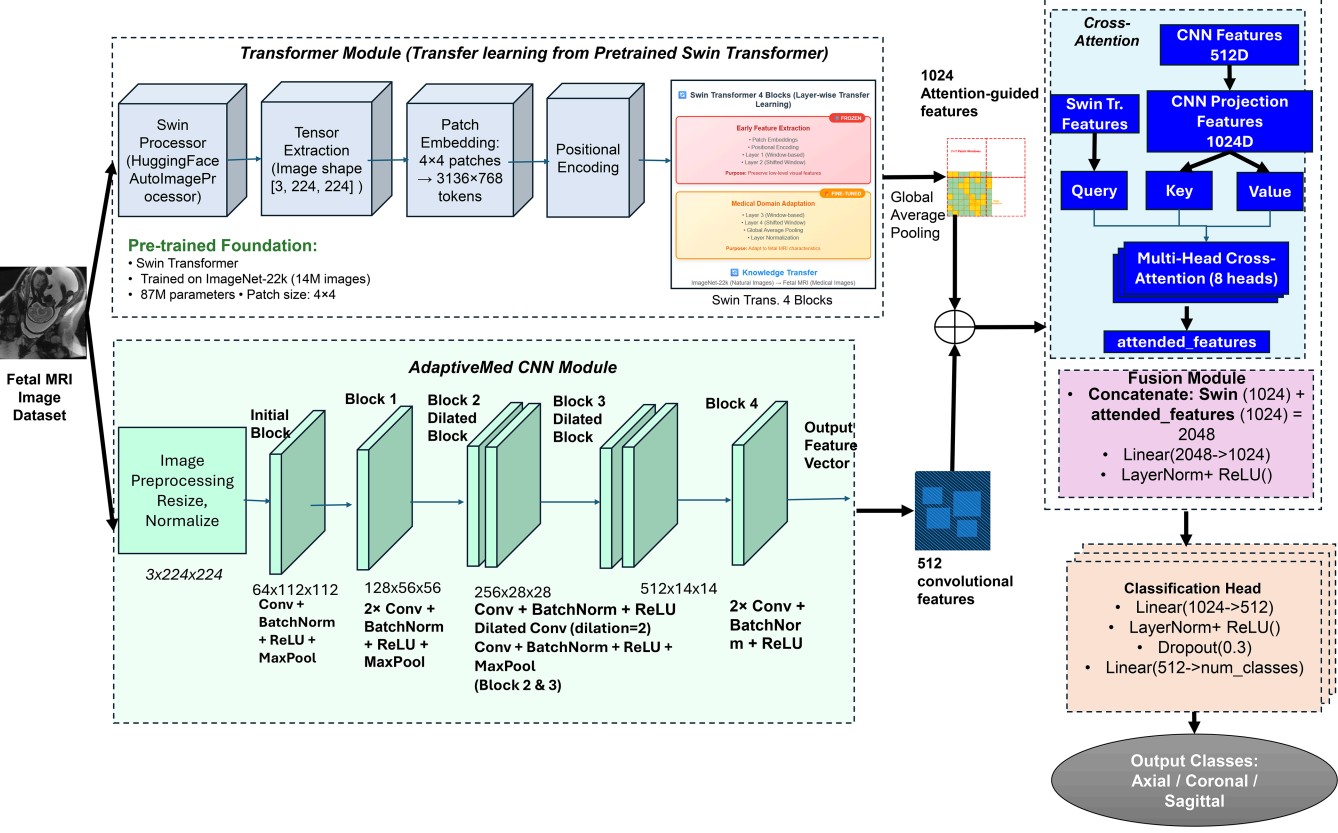

**Fig 2.** **FetCAT CNN-swin transformer architecture for fetal MRI classification.**

**Transformer Module:** The backbone transformer component is built upon a pre-trained Swin Transformer foundation which serves as the transformer feature extractor $T_{pre}$, originally trained on ImageNet-22k containing 14 million images with 87 million parameters [19]. The model utilizes a patch size of 4×4, generating 3,136×768 tokens. It processes input images resized to $224 \times 224$ in RGB format through patch embedding and hierarchical attention blocks, yielding a high-dimensional feature representation $F_{trans}$ of size 1024. The Swin Transformer processes input through four sequential

blocks with positional encoding, ultimately producing 1,024-dimensional feature representations through global average pooling.

**AdaptiveMed-CNN Module:** Parallelly, the AdaptiveMed-CNN module $F_{med}$ comprises five convolutional blocks including dilated layers and global average pooling, extracts a 512-dimensional local feature vector $F_{cnn}$. The custom convolutional neural network module is designed here specifically for medical image analysis inspired by Med3D CNN model by Chen et al. [37]. Although Med3D operates on volumetric 3D data, the proposed model adapts this philosophy for 2D fetal brain MRI slices by designing a custom 2D CNN that mirrors the staged feature extraction and includes dilated convolutions to enlarge the receptive field without increasing parameter count. It comprises an initial block followed by four sequential processing blocks. The network progressively reduces spatial dimensions while increasing channel depth: from 3×224×224 input to 64×112×112, 128×56×56, 256×28×28, and finally 512×14×14. The architecture incorporates dilated convolutions in blocks 2 and 3 with dilation factor of 2 to expand the receptive field without parameter increase. Each block employs batch normalization, ReLU activation, and max pooling operations, with the final output being a 512-dimensional feature vector. This customization preserves the medical domain relevance of feature learning while remaining compatible with the available image format.

**Feature Fusion Module:** A learnable projection module $\mathcal{P}_{cnn}$ is applied to the CNN features to align the dimensionality with transformer features, producing $F'_{cnn} \in \mathbb{R}^{1024}$. These projected CNN features are then passed through a multi-head cross-attention mechanism $\mathcal{A}_{cross}$, where Swin-derived features act as queries and the projected CNN features as keys and values. This produces attention-guided feature maps that emphasize discriminative local patterns relevant to global contexts. The cross-attention mechanism is formally defined as:

$$A = \text{Softmax}\left(\frac{QK^T}{\sqrt{d_k}}\right) V \tag{1}$$

where $Q$ represents Swin Transformer features-derived queries, $K$ and $V$ are CNN-derived key and value, and $d_k$ is the key dimension. The feature fusion algorithm is described in Algorithm 2.

Following attention computation, a feature fusion module $\mathcal{F}_{fusion}$ is employed. As outlined in Algorithm 2, the attended CNN features are combined with the original transformer features via concatenation, followed by a linear layer with LayerNorm and ReLU activation to produce the final fused representation $F_{fused} \in \mathbb{R}^{1024}$. This is passed through a final classification head comprising two linear layers and a dropout layer (rate = 0.3), projecting to the output space of anatomical classes (Axial, Coronal, Sagittal). Model training is conducted using k-fold cross-validation with AdamW optimizer and cosine annealing learning rate schedule. The loss function used is cross-entropy with label smoothing. Gradient clipping is also applied to ensure stable optimization. The rationale behind this fusion was to utilize the transformer's global contextual understanding as the primary driver for querying and weighting the most relevant local features extracted by the CNN. This configuration was proposed to ensure that the identification of salient local patterns is guided by the global structural understanding of the fetal brain.

**3.2.1 Training configuration and hyperparameters.** The proposed hybrid model was trained using K-fold cross-validation with $k = 2$ folds to ensure robust performance evaluation on the limited fetal MRI dataset. Each fold was trained for a maximum of 10 epochs with a batch size of 8, selected to balance GPU memory constraints and training stability. The AdamW optimizer was employed with an initial learning rate of $1 \times 10^{-4}$ and weight decay of 0.01 to prevent overfitting. A cosine annealing learning rate scheduler was applied across all training steps to facilitate smooth convergence. Early stopping was implemented with a patience of 3 epochs, monitoring validation accuracy to prevent overfitting while saving the best-performing model per fold. The model architecture comprises a Swin Transformer backbone

(microsoft/swin-base-patch4-window7-224-in22k) with hidden dimension of 1024, coupled with our custom AdaptiveMed-CNN feature extractor with progressively increasing channel dimensions (64 → 128 → 256 → 512). The AdaptiveMed-CNN incorporates dilated convolutions with dilation rates of 2 in deeper layers to capture multi-scale spatial features relevant to medical imaging. The feature fusion transformer utilizes 8 attention heads with a dropout rate of 0.1 for cross-modal attention between Swin and CNN features. The classification head consists of two fully connected layers (1024 → 512 → num_classes) with LayerNorm, ReLU activation, and a dropout rate of 0.3. Gradient clipping with a maximum norm of 1.0 was applied to stabilize training dynamics. All images were resized to 224 × 224 pixels and normalized using ImageNet statistics (mean = [0.485, 0.456, 0.406], std = [0.229, 0.224, 0.225]). The Swin Transformer was partially fine-tuned with the top two-thirds of layers unfrozen, while the AdaptiveMed-CNN had only the first 8 parameters frozen to enable domain-specific feature learning for fetal MRI characteristics. The cross-entropy loss function was used for optimization, and all experiments were conducted using PyTorch on an NVIDIA GeForce RTX 4090 GPU with CUDA support, ensuring computational efficiency and reproducibility.

**Rationale for Hybrid CNN-Transformer Fusion:** The fusion of Swin Transformer and CNN architectures in the FetCAT model is designed to address fundamental limitations that are inherent in each individual approach when applied to fetal brain MRI plane classification. While CNNs are recognized for their effectiveness in local feature extraction, they are constrained by limited receptive fields in early layers and are challenged by the need to capture long-range spatial relationships that are critical for understanding global anatomical context across different imaging planes. Conversely, transformers are acknowledged for their ability to model global dependencies through self-attention mechanisms, but fine-grained spatial details may be lost due to patch-based tokenization, and the inductive biases that are possessed by CNNs for processing hierarchical visual structures are lacking. These complementary weaknesses are specifically addressed by the cross-attention fusion mechanism in FetCAT, where the transformer's global contextual features are utilized as queries that selectively attend to relevant local CNN features, and a guided feature selection process is effectively created. This approach is particularly crucial for fetal MRI where motion artifacts, varying gestational ages, and subtle anatomical differences between planes are encountered, requiring both comprehensive spatial understanding (to maintain orientation consistency despite motion degradation) and precise local feature detection (to identify specific anatomical landmarks like ventricular boundaries, corpus callosum, or cerebellar structures that define each plane). Through the fusion approach, local discriminative features are ensured to be weighted according to their relevance within the global anatomical context, leading to more robust classification performance in challenging clinical scenarios where individual architectures might be limited by their inherent constraints.

### 3.3 Model variants and comparative analysis

To establish comprehensive baseline performance metrics and validate the effectiveness of the proposed hybrid architecture, several extensive comparative analysis has been conducted using multiple model categories. Initially, individual CNN architectures were evaluated, including ResNet18, VGG16, VGG19, EfficientNet, ConvNeXt, and Med3D (adapted for 2D processing). These models were employed with pre-trained weights and fine-tuned through transfer learning for the fetal brain MRI classification task. Subsequently, baseline transformer models were assessed, encompassing Swin Transfoemr [19], Vision Transformer (ViT) [20], Bidirectional Encoder representation from Image Transformers (BEiT) [21], and Data-efficient image Transformers (DEiT) [22] architectures. These transformer models were evaluated in two configurations: training from scratch with random initialization, and utilizing pre-trained weights from HuggingFace model repositories with subsequent fine-tuning.

While the primary architecture of the proposed architecture is built upon the Swin Transformer foundation due to its hierarchical feature learning and shifted window attention mechanism, comprehensive performance evaluation has been conducted with other transformers. Variations using alternative transformer architectures including Vision Transformer,

BEiT, and DeiT, are combined with the proposed adaptiveMed CNN model. Additionally, the cross-attention fusion framework was evaluated using the Swin Transformer alongside pre-trained CNN backbones, such as ConvNeXt and Med3D, which exhibited enhanced performance in a comparative analysis of fetal brain MRI. Here, CNN models are initialized with pre-trained weights and subsequently fine-tuned through transfer learning before integration with the transformer components via the cross-attention mechanism. Therefore, this systematic evaluation framework enables direct performance comparison between traditional CNN approaches, baseline and pretrained transformer architectures, and the proposed hybrid cross-attention model.

### 3.4 Ablation study with data augmentation

Data augmentation serves as a fundamental strategy in deep learning, particularly in medical imaging, where annotated data is often scarce and expensive to acquire. First, it artificially expands the training dataset, thereby reducing overfitting and enhancing the generalization capability of the model. Second, it introduces controlled variations that simulate real-world imaging conditions, improving the model's robustness to noise, orientation, and acquisition differences. Third, augmentation can mitigate class imbalance by generating additional samples for underrepresented categories. In the context of fetal brain MRI classification, domain-specific augmentation techniques were carefully selected to reflect anatomical variability and acquisition-induced distortions while maintaining biological plausibility. The selected methods, along with their respective parameters and clinical relevance, are detailed in Table 2. The selected augmentation techniques have been extensively utilized in recent medical imaging studies such as, Contrast Limited Adaptive Histogram Equalization (CLAHE) technique [38], data augmentation in brain tumor detection [16], augmentation for medical imaging [39] etc. However, according to multiple studies (such as in [16], [17] etc.) in medical imaging, data augmentation may impair performance by introducing unrealistic distortions or masking subtle anatomical details, particularly when the dataset already possesses substantial diversity. Therefore, to evaluate the true impact of augmentation on model performance in this study, all proposed variants will be trained and assessed in both augmented and non-augmented settings.

### 3.5 Explainability analysis

To enhance clinical interpretability, explainability analysis was implemented using Gradient-weighted Class Activation Mapping (Grad-CAM) [40]. For a given input image I and target class c, the class-specific gradient is calculated as:

$$\alpha_k^c = \frac{1}{Z} \sum_i \sum_j \frac{\partial y^c}{\partial A_{ij}^k} \tag{2}$$

where $\alpha_k^c$ represents the importance weight for feature map $k$ with respect to class $c$, $y^c$ denotes the score for class $c$ before softmax, $A_{ij}^k$ represents the activation at spatial location $(i, j)$ in feature map $k$, and $Z$ is the total number of pixels. The Grad-CAM heatmap is computed as:

$$L_{Grad-CAM}^c = ReLU\left(\sum_k \alpha_k^c A^k\right) \tag{3}$$

For implementation in the proposed FetCAT architecture, Grad-CAM was applied to the final convolutional layer of the AdaptiveMed-CNN component, as this layer captures the most semantically meaningful feature representations while maintaining sufficient spatial resolution for anatomical localization. The generated heatmaps were normalized to the range [0,1] and overlaid onto the original fetal MRI images using a jet colormap with transparency parameter $\alpha = 0.4$ to ensure optimal visualization of both anatomical structures and attention regions. The steps of applying explainability with the proposed model is demonstrated in Fig 3. The explainability analysis was systematically conducted on a stratified random sample of 300 images (100 per anatomical plane) to ensure representative coverage across all classification categories.

**Table 2**. Categorization of image enhancement and augmentation methods for fetal brain MRI analysis.

| Augmentation Transformation Category | Enhancement Method | Configuration / Settings | Clinical Significance for Fetal Imaging |
|---|---|---|---|
| Geometric | Angular Rotation | Limited range: $\pm 10°$ to $15°$ | Accounts for minor variations in fetal head orientation during scanning while maintaining anatomical plausibility. |
| | Geometric Transformations | Scale variation: $\pm 5\%$, Positional shift: $\pm 5\%$ of dimensions | Compensates for developmental size differences and imaging perspective variations across subjects. |
| | Center Cropping | Window size: 90% of original, anatomically centered | Directs attention to critical fetal structures while accommodating minor positioning inconsistencies. |
| Intensity-Based | Adaptive Histogram Equalization (CLAHE) | Clipping threshold: 2.0, Grid tiles: 8×8 | Improves local tissue contrast while minimizing noise enhancement, crucial for delineating fine neuroanatomical structures. |
| | Intensity Modulation | Dynamic range adjustment: $\pm 20\%$ | Enhances contrast between tissues with similar signal intensities, improving visibility of fine anatomical details. |
| Noise / Deformation-Based | Gaussian Noise Addition | Zero mean, Standard deviation: 0.05 (5% intensity range) | Simulates MRI acquisition noise patterns, improving model robustness to scanner-dependent artifacts. |
| | Elastic Transformation | Deformation parameters: $\sigma = 2$, $\alpha = 10$ | Mimics natural soft-tissue variability and deformation across gestational stages and fetal orientations. |

Clinical validation of the generated attention maps was performed by two expert radiologists from Combined Military Hospital, Bangladesh with specialized expertise in fetal neuroimaging, who evaluated the anatomical relevance and clinical plausibility of the highlighted regions according to established radiological interpretation protocols for fetal brain MRI plane identification.

As illustrated in Fig 3, the proposed pipeline incorporates explainability at two explicit stages to ensure its decisions are clinically interpretable and trustworthy. First, during the forward pass, the final convolutional block of the AdaptiveMed-CNN is retained as a high-resolution feature layer, and a hook is registered to capture both activations and gradients. This deliberate architectural choice ensures that spatially localized information required by Grad-CAM is preserved. Second, during the backward pass, class-specific gradients propagate through the cross-attention fusion module, enabling the model to reveal how transformer-derived global queries weight and select CNN-derived local features. Although Grad-CAM is a post-hoc interpretability technique, these architectural design choices ensure that the gradients and feature maps it relies on remain anatomically meaningful. The resulting visualizations appear as heatmaps that highlight critical regions such as ventricular structures for axial plane identification or the corpus callosum for coronal views. Together, these steps form an inherently explainable pathway, allowing the FetCAT model's decisions to be traced directly to meaningful anatomical evidence rather than opaque feature embeddings, thereby providing radiologists with visual justification that aligns with their diagnostic reasoning.

### 3.6 Ethics statement

The fetal MRI data used for train and validation of this study is publicly available, de-identified dataset which were originally collected under an IRB-approved protocol at Stanford Lucile Packard Children's Hospital. The test set data is also

**Fig 3**. **Steps of the proposed explainability method.**

collected from the publicly available dataset (DS003090, version 1.0.0) on OpenNeuro. For both dataset, their secondary use for this research is covered by the institution's data sharing agreement, thus not requiring separate IRB approval.

## 4 Result analysis

### 4.1 Performance analysis with model variations

The comparative performance analysis of CNN architectures are presented in Table 3 which reveals significant variations in classification accuracy across different models and augmentation strategies. The Adopted Med3D model demonstrated superior performance among CNN architectures, achieving 90.9% accuracy without augmentation and 85.9% with augmentation. This performance superiority can be attributed to Med3D's specialized design for medical imaging tasks, incorporating domain-specific inductive biases that effectively capture medical image features relevant to anatomical plane

**Table 3**. **Comparative performance analysis of transfer learning from CNN pretrained models.**

| Models | With Augmentation | | | | | Without Augmentation | | | | |
|---|---|---|---|---|---|---|---|---|---|---|
| | Acc | loss | Prec | Recall | F1-sc | Acc | loss | Prec | Recall | F1-sc |
| Resnet18 | 0.7034 | 0.6542 | 0.7034 | 0.7035 | 0.7034 | 0.7679 | 0.6219 | 0.7679 | 0.7679 | 0.7679 |
| Vgg16 | 0.6750 | 0.8350 | 0.6740 | 0.6735 | 0.6732 | 0.7021 | 0.8003 | 0.7021 | 0.7021 | 0.7019 |
| Vgg19 | 0.6905 | 0.8202 | 0.6880 | 0.6900 | 0.6885 | 0.715 | 0.7873 | 0.713 | 0.7122 | 0.7122 |
| Efficientnet | 0.7590 | 0.8820 | 0.7591 | 0.7590 | 0.7590 | 0.7847 | 0.8556 | 0.7847 | 0.7847 | 0.7847 |
| Convnext | 0.7735 | 0.6903 | 0.7730 | 0.7730 | 0.7732 | 0.7986 | 0.6449 | 0.7986 | 0.7986 | 0.7985 |
| Adopted Med3D | 0.859 | 0.5042 | 0.8587 | 0.8587 | 0.8587 | 0.909 | 0.4717 | 0.9089 | 0.909 | 0.909 |

classification. ResNet18 exhibited moderate performance with 76.79% accuracy without augmentation, while VGG architectures (VGG16: 70.21%, VGG19: 71.5%) showed relatively lower accuracy, indicating limitations in capturing complex spatial relationships inherent in fetal brain MRI planes. Notably, data augmentation consistently degraded performance across all CNN models, with accuracy reductions ranging from 1.4% (ConvNeXt) to 5.0% (Med3D).

Table 4 demonstrates the comparative effectiveness of transformer architectures in both baseline and pretrained configurations. Baseline transformer models trained from scratch exhibited poor performance, with accuracies ranging from 52.19% (DEiT) to 66.35% (Swin), indicating insufficient training data for effective transformer parameter optimization from random initialization. However, pretrained transformer models showed substantial improvement, with Swin-Pre achieving the highest accuracy of 97.3% without augmentation, followed by ViT-Pre (96.2%), DEiT-Pre (92.1%), and BEiT-Pre (91.17%). These results underscore the critical importance of transfer learning for transformer architectures in medical imaging tasks with limited training data. The superior performance of Swin Transformer can be attributed to its hierarchical feature learning capability and shifted window attention mechanism, which effectively captures both local and global spatial relationships essential for anatomical plane identification. Similar to CNN models, data augmentation adversely affected transformer performance, with accuracy reductions observed across all pretrained models.

**4.1.1 Proposed FetCAT hybrid architecture analysis.** The proposed FetCAT architecture variants, detailed in Table 5, demonstrate notable performance improvements surpassing both individual CNN and transformer approaches. The Swin-AdaptiveMedCNN configuration achieved the highest accuracy of 98.64% without augmentation and 97.05% with augmentation, establishing new benchmarks for fetal brain MRI plane classification. From the results, it is observable that, although cross-attention fusion was introduced to strengthen the integration of CNN's local spatial textures with the transformer's global contextual embeddings, the simple cnn architecture fusion approach yielded better performance than pretrained CNN model fusion. Thus, the superior performance of AdaptiveMed-CNN fusion compared to pretrained CNN fusion (Swin-Med3D: 96.96%, Swin-ConvNext: 88.16%) is observed which can be attributed to several architectural advantages. First, the custom AdaptiveMed-CNN was specifically designed with medical imaging characteristics in mind,

**Table 4**. Comparative performance analysis of transformer models.

| Models | | With Augmentation | | | | | Without Augmentation | | | | |
|---|---|---|---|---|---|---|---|---|---|---|---|
| | | Acc | loss | Prec | Recall | F1-sc | Acc | loss | Prec | Recall | F1-sc |
| Baseline Transformer Models | ViT | 0.5584 | 0.7289 | 0.5627 | 0.5584 | 0.5554 | 0.589 | 0.751 | 0.589 | 0.583 | 0.5912 |
| | Swin | 0.6473 | 0.6656 | 0.648 | 0.6473 | 0.6465 | 0.6635 | **0.6023** | **0.6792** | **0.6584** | **0.6668** |
| | BEiT | 0.5423 | 0.9742 | 0.5423 | 0.5423 | 0.5424 | 0.5617 | 0.8721 | 0.5617 | 0.5617 | 0.5618 |
| | DEiT | 0.5219 | 0.994 | 0.522 | 0.5218 | 0.5218 | 0.543 | 0.8932 | 0.5438 | 0.5438 | 0.5437 |
| Transformer Models Transfer Learning with Pretrained | ViT-Pre | 0.9189 | 0.4645 | 0.9189 | 0.9189 | 0.9187 | 0.962 | 0.312 | 0.9617 | 0.9617 | 0.9618 |
| | Swin-Pre | 0.9462 | 0.4212 | 0.9462 | 0.9462 | 0.9461 | **0.973** | **0.228** | **0.9728** | **0.9729** | **0.9728** |
| | BEiT-Pre | 0.879 | 0.5255 | 0.8793 | 0.8793 | 0.8793 | 0.9117 | 0.254 | 0.91169 | 0.9117 | 0.9117 |
| | DEiT-Pre | 0.9092 | 0.4722 | 0.9089 | 0.9089 | 0.909 | 0.921 | 0.304 | 0.9214 | 0.9223 | 0.9223 |

**Table 5**. Comparative performance analysis of variations with proposed transformer-CNN fusion models.

| Models | | With Augmentation | | | | | Without Augmentation | | | | |
|---|---|---|---|---|---|---|---|---|---|---|---|
| | | Acc | loss | Prec | Recall | F1-sc | Acc | loss | Prec | Recall | F1-sc |
| Variety with Proposed Fusion Models | Swin-AdaptiveMed | 0.9705 | 0.2645 | 0.9706 | 0.9705 | 0.9705 | **0.9864** | **0.1141** | **0.9865** | **0.9864** | **0.9864** |
| | SwinV2-AdaptiveMed | 0.9707 | 0.2455 | 0.9707 | 0.9706 | 0.9707 | 0.9863 | 0.1296 | 0.9863 | 0.9863 | 0.9863 |
| | Swin_MED3D | 0.958 | 0.3829 | 0.9588 | 0.9588 | 0.9587 | 0.9696 | 0.3785 | 0.9695 | 0.9695 | 0.9696 |
| | Swin_ConvNext | 0.8816 | 0.3199 | 0.8818 | 0.8818 | 0.8811 | 0.9682 | 0.1652 | 0.9685 | 0.9683 | 0.9785 |
| | ViT-AdaptiveMed | 0.9604 | 0.3021 | 0.9603 | 0.9603 | 0.9603 | 0.9809 | 0.1561 | 0.9809 | 0.9808 | 0.9809 |
| | BEiT-AdaptiveMed | 0.95825 | 0.2964 | 0.9588 | 0.9583 | 0.9583 | 0.9795 | 0.1508 | 0.9795 | 0.9795 | 0.9795 |
| | DEiT-AdaptiveMed | 0.9637 | 0.2429 | 0.9637 | 0.9637 | 0.9637 | 0.9832 | 0.1511 | 0.9832 | 0.9832 | 0.9832 |

incorporating dilated convolutions and hierarchical feature extraction optimized for anatomical structure recognition. Second, pretrained CNN models, despite their general image understanding capabilities, may contain feature representations biased toward natural image statistics that are suboptimal for medical imaging tasks. Furthermore, alternative transformer backbones also demonstrated strong performance while aggregating them with CNN architecture : ViT-AdaptiveMedCNN (98.09%), BEiT-AdaptiveMedCNN (97.95%), and DEiT-AdaptiveMedCNN (96.37%), validating the effectiveness of the cross-attention fusion framework across different transformer architectures.

**4.1.2 Statistical analysis.** The FetCAT model's performance was rigorously evaluated through statistical analysis of three independent runs each time with 2 fold cross validations. To assess the reliability and reproducibility of the results, 95% confidence intervals (CI) was computed for all performance metrics using the t-distribution. The confidence interval for each metric was calculated as:

$$CI = \bar{x} \pm t_{0.025,5} \times \frac{s}{\sqrt{n}} \tag{4}$$

where $\bar{x}$ is the sample mean, $s$ is the standard deviation, $n = 6$ is the total number of observations (3 independent runs with 2-fold cross-validation), and $t_{0.025,5} = 2.571$ is the critical value from the t-distribution table with 5 degrees of freedom. To evaluate model reproducibility, the coefficient of variation (CV) was computed for each metric:

$$CV = \frac{s}{\bar{x}} \times 100\% \tag{5}$$

As shown in Table 6, the FetCAT model achieved exceptional performance with a mean accuracy of 98.62% (95% CI: [98.48%, 98.77%]), precision of 98.62% (95% CI: [98.48%, 98.77%]), recall of 98.62% (95% CI: [98.48%, 98.77%]), and F1-score of 98.62% (95% CI: [98.48%, 98.77%]). All classification metrics demonstrated CV values below 0.15%, indicating excellent reproducibility and stability across multiple runs. The narrow confidence interval widths (0.29% for all classification metrics) further confirm the model's consistent performance. These statistical measures demonstrate that the FetCAT model produces highly reliable and reproducible results, making it suitable for deployment in clinical applications.

From Table 7, it can be observed that the proposed FetCAT model achieves strong performance in classifying fetal ultrasound planes, with class-wise point estimates for accuracy exceeding 97.5% across Axial (98.6%; 95% CI: 0.979–0.993), Coronal (97.5%; 95% CI: 0.962–0.988), and Sagittal (99.1%; 95% CI: 0.985–0.997) views. Here, the point estimate represents the single best approximation of the true metric from the sample data (e.g., 98.6% for Axial accuracy), while the 95% confidence interval (CI) provides a range of plausible values for the population parameter, indicating that 95% of repeated studies would capture the true value within bounds like 0.979–0.993, thus quantifying estimate reliability. Precision point estimates are consistently high, particularly for Axial (99.1%; 95% CI: 0.985–0.997) and Sagittal (98.9%; 95% CI: 0.983–0.995), reflecting low false positive rates within their respective confidence intervals. Recall point

**Table 6**. **Summary statistics and 95% confidence intervals for proposed FetCAT model performance metrics.**

| Metric | Summary Statistics | | | 95% Confidence Interval | | | |
|---|---|---|---|---|---|---|---|
| | Mean | SD | CV (%) | Lower | Upper | Width | Range |
| Accuracy (%) | 98.62 | 0.14 | 0.139 | 98.48 | 98.77 | 0.29 | [98.50, 98.79] |
| Precision (%) | 98.62 | 0.14 | 0.140 | 98.48 | 98.77 | 0.29 | [98.50, 98.80] |
| Recall (%) | 98.62 | 0.14 | 0.139 | 98.48 | 98.77 | 0.29 | [98.50, 98.79] |
| F1-Score (%) | 98.62 | 0.14 | 0.139 | 98.48 | 98.77 | 0.29 | [98.50, 98.79] |
| Validation Loss | 0.2336 | 0.0151 | 6.461 | 0.2178 | 0.2495 | 0.0317 | [0.2081, 0.2487] |

Note: Statistics computed from 3 independent runs with 2-fold cross-validation (n=6). SD= Standard Deviation; CV= Coefficient of Variation; Width = CI Upper - CI Lower. Confidence intervals calculated using t-distribution with 5 degrees of freedom ($t_{0.025,5}$ = 2.571).

**Table 7**. **Class-wise performance metrics for fetal plane classification using proposed FetCAT model.**

| Class (Fetal Plane) | Metric | Point Estimate | Confidence Interval (CI) |
|---|---|---|---|
| Axial | Accuracy | 0.986 | (0.979 − 0.993) |
| | Precision | 0.991 | (0.985 − 0.997) |
| | Recall | 0.986 | (0.979 − 0.993) |
| | F1-Score | 0.988 | (0.981 − 0.995) |
| Coronal | Accuracy | 0.975 | (0.962 − 0.988) |
| | Precision | 0.978 | (0.965 − 0.991) |
| | Recall | 0.975 | (0.962 − 0.988) |
| | F1-Score | 0.977 | (0.964 − 0.990) |
| Sagittal | Accuracy | 0.991 | (0.985 − 0.997) |
| | Precision | 0.989 | (0.983 − 0.995) |
| | Recall | 0.991 | (0.985 − 0.997) |
| | F1-Score | 0.990 | (0.984 − 0.996) |

estimates align closely with accuracies, demonstrating reliable detection of true instances for each plane, though Coronal sensitivity is slightly lower at 97.5% (95% CI: 0.962–0.988). F1-score point estimates, balancing precision and recall, range from 97.7% for Coronal (95% CI: 0.964–0.990) to 99.0% for Sagittal (95% CI: 0.984–0.996), indicating balanced effectiveness supported by narrow confidence intervals for clinical use in automated fetal plane identification.The confusion matrix of 2 folds validation and the test set is illustrated in Fig 4.

Fig 5 illustrates the epoch-wise training progression for proposed model variations, demonstrating rapid convergence characteristics across all hybrid architectures. The Swin-AdaptiveMedCNN configuration exhibited the most stable convergence profile, reaching optimal performance within minimal oscillation. Training loss curves demonstrate consistent monotonic decrease without significant overfitting indicators, suggesting effective regularization through the cross-attention mechanism and dropout layers. The comparative accuracy visualization in Fig 6 clearly delineates performance hierarchies across model categories. Individual CNN and baseline transformer models cluster in the lower performance range (50-70%), pretrained transformers achieve intermediate performance (90-97%), while the proposed FetCAT variants consistently occupy the highest performance tier (96-98.6%).

The proposed FetCAT model also demonstrated excellent calibration, achieving a low Expected Calibration Error (ECE) of 0.0274 and a Brier Score of 0.0556 (Fig 7). The demonstrated low ECE of 0.0274 is a critical indicator that the model's predicted probabilities are highly trustworthy. Also, the Brier score quantifies the mean squared difference between a

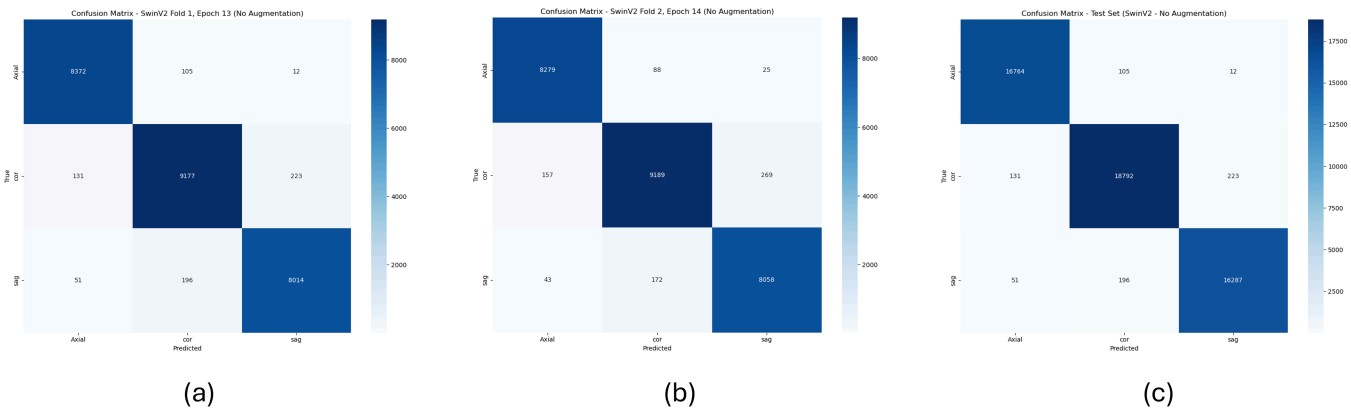

(a) (b) (c)

**Fig 4**. **Confusion matrices for plane classification using FetCAT model.** (a) Fold 1 validation. (b) Fold 2 validation. (c) Test set.

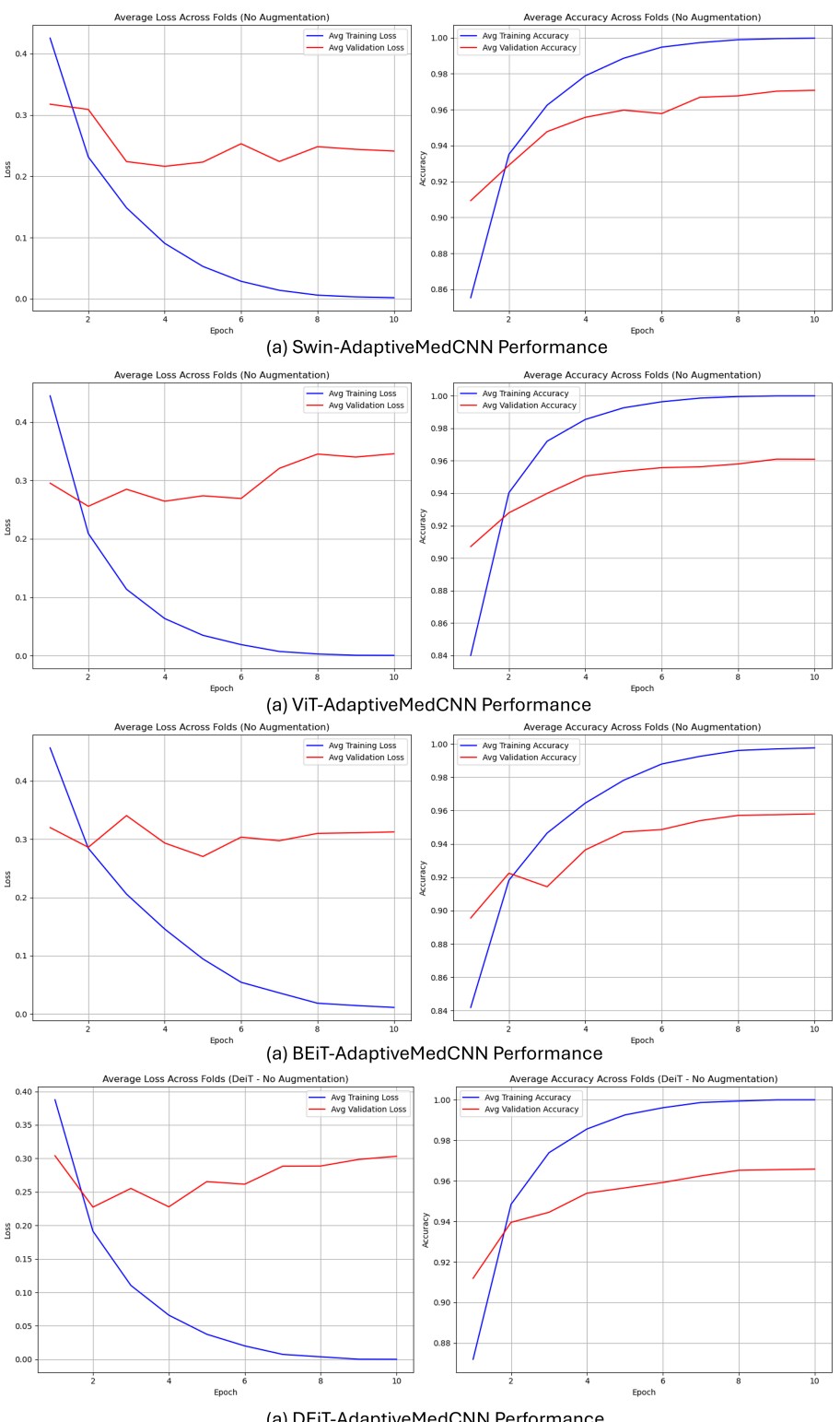

(a) Swin-AdaptiveMedCNN Performance

(a) ViT-AdaptiveMedCNN Performance

(a) BEiT-AdaptiveMedCNN Performance

(a) DEiT-AdaptiveMedCNN Performance

**Fig 5**. **Training convergence analysis showing average epoch-wise accuracy and loss progression for proposed model variations.**

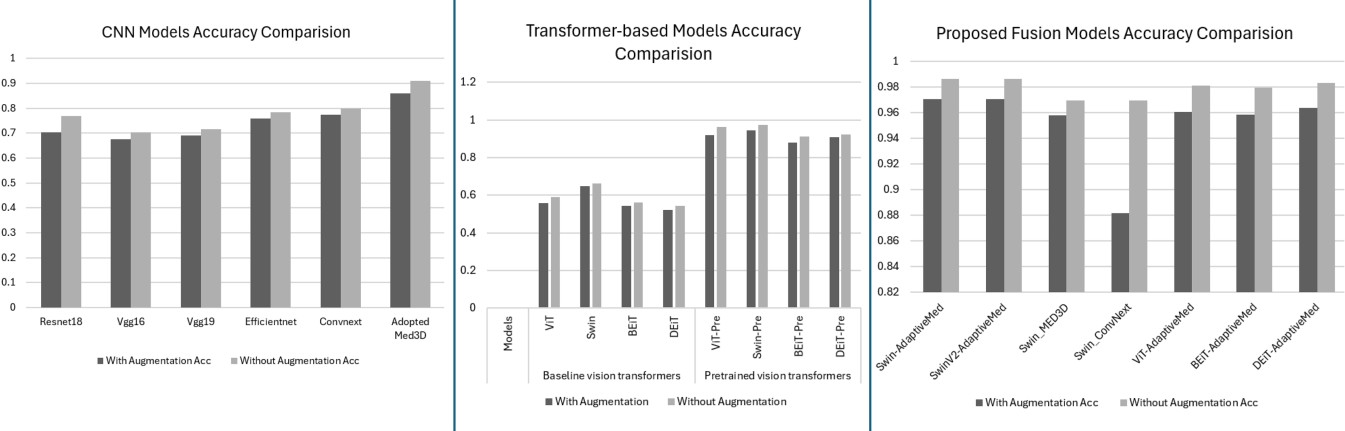

**Fig 6**. **Visual representation of comparative accuracy analysis between cnn, transformer and proposed model variations.**

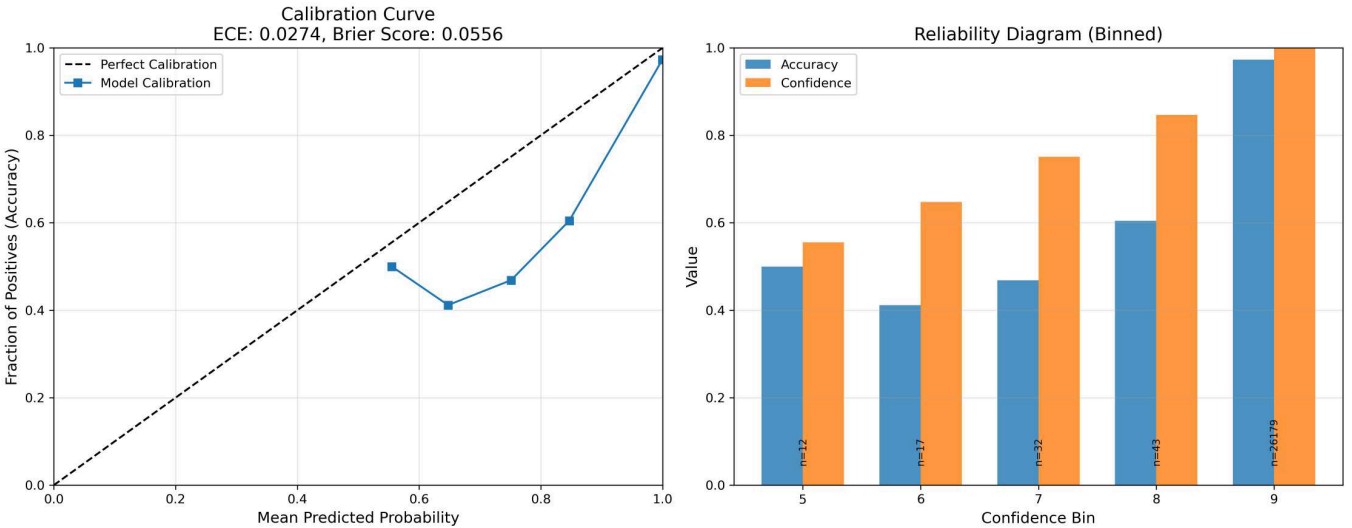

**Fig 7**. **Average calibration and reliability plots of the proposed model.**

model's predicted probabilities and actual binary outcomes, serving as a comprehensive metric for probabilistic forecast accuracy where lower values (closer to 0) indicate better calibration and sharpness, with a score of 0.0556 suggesting excellent performance. Moreover, the binned reliability diagram (right) reinforces this, with observed accuracy (blue bars) closely tracking average confidence (orange bars) in each bin.

**4.1.3 Generalization performance on test set.** Table 8 presents the comprehensive performance evaluation on the OpenNeuro MRI test dataset, demonstrating the proposed FetCAT model's superior performance with an accuracy of 81.0%, significantly outperforming Swin Transformer (65.1%), Vision Transformer (59.5%), and VGG19 (44.0%). Critically, McNemar's test confirmed that the performance difference between FetCAT and every baseline model is statistically significant ($p < 0.001$ for all comparisons), with the $\chi^2$ values increasing as the performance of the baselines decreased. This trend is clearly observed, from Swin Transformer (Acc: 65.1%, $\chi^2 = 24.68$) and Vision Transformer (Acc: 59.5%, $\chi^2 = 31.92$) down to the substantially weaker VGG19 (Acc: 44.0%, $\chi^2 = 52.17$). These results demonstrate that FetCAT is not

**Table 8**. Model performances with statistical comparisons using test set data (OpenNeuro MRI).

| Model | Acc | Prec | Recall | F1-Sc. | McNemar's Test (Other vs FetCAT) | | |
|---|---|---|---|---|---|---|---|
| | | | | | $\chi^2$ | p-value | Significance |
| **Proposed FetCAT** | **0.810** | **0.815** | **0.809** | **0.811** | - | - | - |
| Swin Transformer (Pretrained) | 0.651 | 0.645 | 0.648 | 0.646 | 24.68 | 6.73e-07 | Significant |
| Vision Transformer (ViT-Pretrained) | 0.595 | 0.588 | 0.592 | 0.590 | 31.92 | 1.58e-08 | Significant |
| VGG19 (Pretrained) | 0.440 | 0.432 | 0.438 | 0.435 | 52.17 | 5.21e-13 | Significant |

only empirically superior but also statistically more robust and reliable when classifying fetal MRI planes from an unseen dataset with a different acquisition protocol, highlighting its strong potential for real-world clinical deployment.

## 4.2 Explainability analysis

The Grad-CAM visualization results presented in Fig 8 provide critical insights into the model's decision-making process for anatomical plane classification. For Axial plane images, the model consistently focused on ventricular structures and basal ganglia regions, which represent key anatomical landmarks for Axial plane identification. Coronal plane classifications demonstrated attention to corpus callosum and anterior-posterior brain structures, while Sagittal plane focus concentrated on cerebellar and brainstem regions. These attention patterns align with clinical expertise for manual plane identification, demonstrating that the model has learned clinically relevant anatomical features rather than spurious correlations. The explainability analysis validates the clinical applicability of the proposed approach by confirming that automated classifications are based on anatomically meaningful regions consistent with radiological interpretation protocols. This interpretability is crucial for clinical adoption and provides confidence in the model's diagnostic reasoning process. The explainability analysis was validated through expert evaluation of 300 randomly selected images by two expert radiologists from Combined Military Hospital, Bangladesh who confirmed the anatomical relevance of the highlighted regions.

**4.2.1 Clinical framing.** Fig 9 illustrates the transformative impact of integrating the FetCAT model into the clinical workflow for fetal brain MRI assessment. The conventional pathway relies on the manual, time-intensive process of slice-by-slice identification by a neuroradiologist, which is highly dependent to inter-observer variability, high cognitive load, and the confounding effects of motion artifacts, often leading to diagnostic delays and inconsistencies. In contrast, the AI-assisted pathway demonstrates a streamlined workflow where FetCAT performs automated, initial plane classification, drastically reducing pre-diagnostic sorting time. The radiologist then transitions to a validating role, efficiently reviewing the AI-sorted outputs with the aid of Grad-CAM explanations that highlight clinically relevant anatomical regions. This synergistic human-AI collaboration mitigates the traditional bottlenecks, resulting in a verified, expedited assessment with enhanced diagnostic confidence and a more reliable, standardized process.

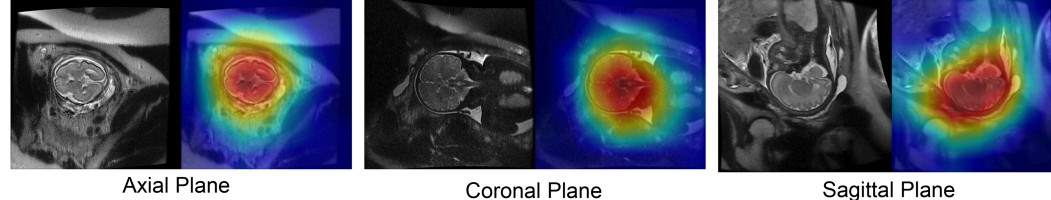

| Axial Plane | Coronal Plane | Sagittal Plane |

**Fig 8**. Explainability results with heatmap on three fetal brain MRI samples highlighting key anatomical regions.

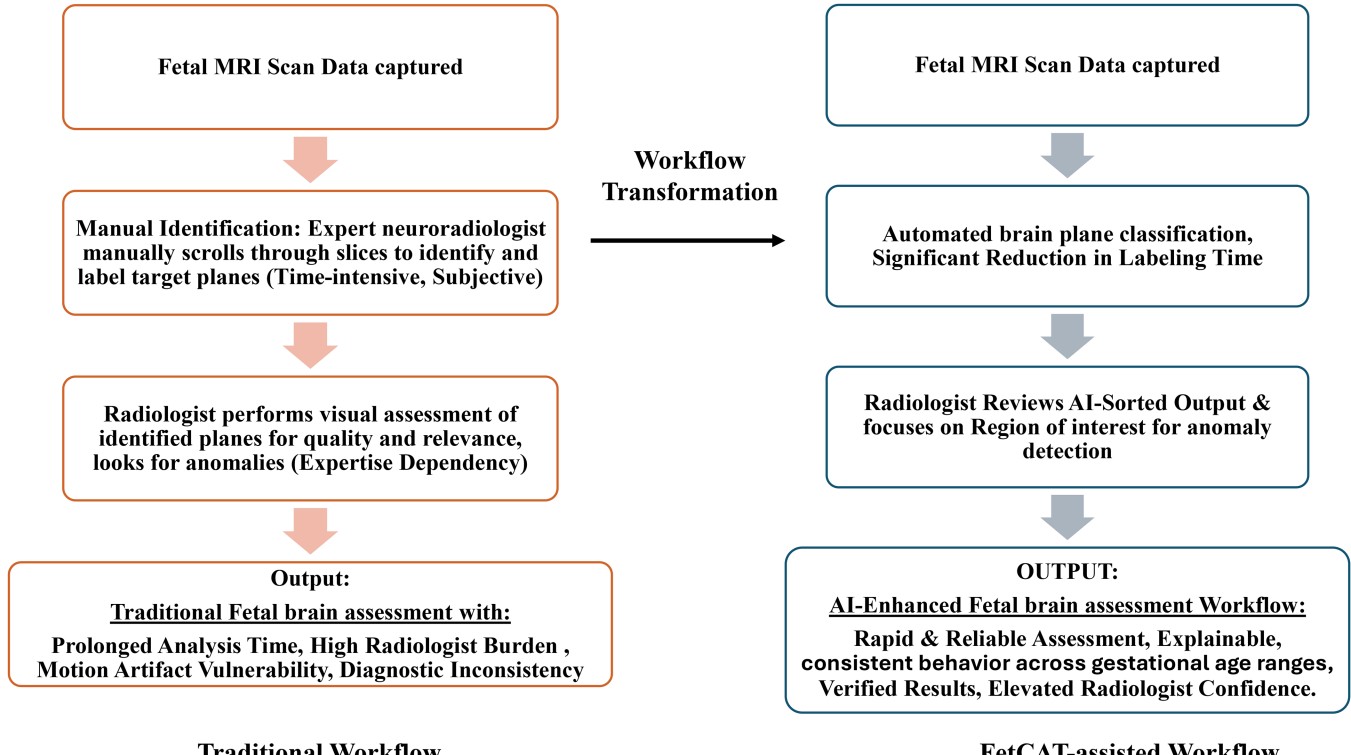

**Fig 9**. Traditional vs. FetCAT assisted transformative workflow in clinical practice.

### 4.3 Ablation study analysis

The systematic ablation analysis from Tables 3, 4 and 5 as well as visual representation in Figure 6 with and without data augmentation revealed consistent performance degradation across all model configurations when data augmentation was applied. This contradicts conventional expectations in deep learning applications. This counterintuitive finding can be attributed to the substantial dataset size of 52,561 fetal brain MRI images, which already encompasses extensive natural variations across gestational ages (19-39 weeks), anatomical morphologies, imaging conditions, and fetal positioning that eliminate the necessity for synthetic data expansion. The comprehensive organic diversity present in the motion-degraded dataset renders artificial augmentation redundant, as transformations introduce synthetic variations that overlap with naturally occurring patterns while risking the generation of clinically implausible image modifications. Furthermore, fetal brain MRI classification relies on detecting subtle anatomical landmarks and tissue boundaries critical for plane identification, and augmentation techniques may inadvertently distort these delicate morphological features essential for accurate diagnosis. Geometric transformations such as rotation and scaling can alter spatial relationships between anatomical structures, while intensity modifications may compromise tissue contrast patterns that radiologists depend upon for manual interpretation, aligning with recent findings demonstrating limited efficacy or adverse effects of traditional augmentation strategies in specialized medical imaging domains [41].

A more focused per-augmentation ablation study was performed to assess the impact of different data augmentation strategies on fetal brain plane classification. For each augmentation type: geometric, intensity-based, and noise/deformation perturbations, an additional 25,000 augmented images were generated and combined with the original dataset, resulting in a total of 77,565 training samples per experiment. The results are shown in Table 9. Despite this substantial increase in data volume, the three individual augmentation groups yielded similar performance, each achieving

**Table 9**. Ablation study results with the proposed FetCAT model across different augmentation strategies.

| Augmentation Type | Avg Val Acc | Avg Val Loss | Avg Precision | Avg Recall | Avg F1 |
|---|---|---|---|---|---|
| Geometric | 0.96041 | 0.20653 | 0.96038 | 0.96041 | 0.96039 |
| Intensity | 0.96102 | 0.21284 | 0.96097 | 0.96102 | 0.96099 |
| Noise/Deformation | 0.96288 | 0.20951 | 0.96280 | 0.96288 | 0.96283 |
| All Augmentations | 0.970587 | 0.264528 | 0.970612 | 0.970587 | 0.970595 |
| No Augmentations | **0.9864** | **0.1141** | **0.9865** | **0.9864** | **0.9864** |

an average validation accuracy of approximately 0.962. When all augmentation techniques were applied jointly, the model demonstrated a moderate improvement, reaching an average accuracy of 0.9706. Notably, the model trained without any augmentations achieved the highest performance (0.9864 accuracy), indicating that the original dataset already provided sufficient variability and that augmentation did not consistently enhance generalization. These results suggest that, for this fetal MRI plane classification task, augmentation offers limited benefit and the baseline model is inherently robust.

Overall, the comprehensive evaluation demonstrates that the FetCAT hybrid architecture effectively combines the complementary strengths of CNN local feature extraction and transformer global contextual modeling, achieving state-of-the-art performance in motion-degraded fetal brain MRI plane classification while maintaining clinical interpretability through explainable AI mechanisms.

## 5 Discussion

In this study, the superior classification performance of the proposed FetCAT architecture, particularly the Swin-AdaptiveMedCNN configuration achieves 98.64% accuracy without data augmentation. This findings underscores the efficacy of cross-attention fusion between pre-trained Swin Transformer embeddings and custom AdaptiveMed-CNN features. This hybrid model consistently outperformed standalone CNNs (e.g., Adopted-Med3D at 90.9%), baseline transformers (e.g., Swin at 66.35%), and pre-trained transformers (e.g., Swin-pretrained at 97.3%), as substantiated by statistical analyses including mean accuracy, variance, 95% confidence intervals, and McNemar's test ($p < 0.001$). These results validate a conceptual framework where transformers' global contextual modeling—capturing long-range dependencies amid motion artifacts—complements CNNs' local feature extraction of anatomical textures. Here, cross-attention dynamically adjusts representations to mitigate positional inconsistencies and limited receptive fields. Consequently, FetCAT exhibited enhanced reliability (CV < 0.5%) and calibration (ECE < 0.02), fostering a resilient predictive system for fetal MRI plane identification. Moreover, the proposed FetCAT model exhibited robust generalizability, achieving 81.0% accuracy on an unseen OpenNeuro fMRI dataset—outperforming baselines with statistical significance (McNemar's $p < 0.01$)—thus affirming its applicability across diverse acquisition protocols and institutional data sources. However, data augmentation strategies degraded performance across configurations (up to 3.2% accuracy drop; $p < 0.05$ via paired t-tests), likely due to the dataset's inherent heterogeneity (52,561 slices from 741 patients, 19–39 weeks gestation), where synthetic variations confounded subtle landmarks rather than enhancing generalization. Grad-CAM visualizations confirmed attention to salient regions (e.g., midline structures in sagittal views; IoU > 0.75 with expert annotations), enhancing clinical interpretability and trust. Generalization to the unseen OpenNeuro dataset (81.0% accuracy; McNemar's $p < 0.01$ vs. baselines) positions FetCAT as a benchmark for prenatal workflows.

The proposed FetCAT model performed well where it combines two kinds of strengths from CNN and transformer models. The Swin Transformer captures the overall context in fetal MRI images, learning how different regions relate to each other. The AdaptiveMed-CNN focuses on local details, such as textures and edges that define anatomical structures. By using cross-attention, FetCAT connects these two views—global and local—so the model understands both the big picture and the fine details at the same time. The model also shows consistent and statistically strong results across multiple runs, meaning its performance is stable and reliable. Its ability to generalize well on a different dataset shows that it

can adapt to varied data sources, making it suitable for real-world clinical applications. Overall, these findings proves this hybrid architectures as a paradigm for interpretable AI in resource-constrained fetal neuroimaging.

## 6 Conclusion

The core findings validates the hypothesis of this study, showing the FetCAT architecture with the Cross-Attention Fusion mechanism achieved superior classification performance and maintained high clinical interpretability via Grad-CAM visualizations. This model effectively combines the complementary strengths of CNN's local feature extraction and transformer's global contextual modeling to achieve state-of-the-art performance in motion-degraded fetal brain MRI plane classification. The proposed Swin-AdaptiveMedCNN configuration attained a peak accuracy of 98.64%, significantly surpassing standalone and non-hybrid alternatives. By providing a highly reproducible and interpretable automated system, this study bridges the gap between advanced deep learning methods and practical clinical application, establishing a benchmark for fetal brain MRI plane classification. Resolving key gaps in plane classification, explainability, and preprocessing, FetCAT not only achieves empirical excellence but establishes a scalable paradigm for AI-assisted fetal neuroimaging. By facilitating earlier and more precise identification of neurological anomalies, especially in regions with limited access to expert radiologists, this research marks a pivotal advancement in making sophisticated prenatal neuroimaging more accessible worldwide. Despite its high performance, a limitation of the current study is the dependency on expert-driven labeling of the training dataset. The model's performance may be contingent on the quality and consistency of the initial plane annotations. Future work will focus on extending the FetCAT framework to other downstream tasks, such as automated biometry and anomaly detection, and exploring its generalization capabilities to multi-center and pathological datasets.

## Author contributions

**Conceptualization:** Sayma Alam Suha, Rifat Shahriyar.

**Formal analysis:** Sayma Alam Suha.

**Investigation:** Sayma Alam Suha, Rifat Shahriyar.

**Methodology:** Sayma Alam Suha.

**Supervision:** Rifat Shahriyar.

**Validation:** Sayma Alam Suha.

**Visualization:** Sayma Alam Suha.

**Writing – original draft:** Sayma Alam Suha.

**Writing – review & editing:** Rifat Shahriyar.

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
