## [Decision Letter · Decision Letter 0]

20 Oct 2025

PONE-D-25-47910FetCAT: Cross-Attention Fusion of Transformer-CNN Architecture for Fetal Brain Plane Classification with Explainability using Motion-degraded MRIPLOS ONE

Dear Dr. Suha,

Thank you for submitting your manuscript to PLOS ONE. After careful consideration, we feel that it has merit but does not fully meet PLOS ONE’s publication criteria as it currently stands. Therefore, we invite you to submit a revised version of the manuscript that addresses the points raised during the review process.

**ACADEMIC EDITOR:**

Address all the remarks raised by the three reviewersSpecify what kind of data you employed in your trialsHighlight the steps of the proposed methods that are "explainable".

We look forward to receiving your revised manuscript.

Kind regards,

Alessandro Bruno, Ph.D.

Academic Editor

PLOS ONE

Additional Editor Comments:

Dear Authors,

Your paper reveals a certain level of depth on the topic of interest.

However, you need to address some weak points that two reviewers have raised.

Please specify the type of data you adopted in your experimental trials. Also, you may want to push forward the bar of explainability and provide more methodological details as requested by one of the three reviewers.

Do your best to answer all comments and remarks from reviewers pointwise.

I recommed your manuscript for a Major Revision round.

Kind regards,

A.B.

Reviewers' comments:

Reviewer's Responses to Questions

**Comments to the Author**

1. Is the manuscript technically sound, and do the data support the conclusions?

Reviewer #1: Yes

Reviewer #2: Partly

Reviewer #3: Yes

2. Has the statistical analysis been performed appropriately and rigorously?

Reviewer #1: No

Reviewer #2: I Don't Know

Reviewer #3: Yes

3. Have the authors made all data underlying the findings in their manuscript fully available?

Reviewer #1: Yes

Reviewer #2: No

Reviewer #3: No

4. Is the manuscript presented in an intelligible fashion and written in standard English?

Reviewer #1: Yes

Reviewer #2: Yes

Reviewer #3: Yes

5. Review Comments to the Author

Reviewer #1: The authors have presented a well-written manuscript highlighting a novel hybrid architecture that integrates a pre-trained Swin Transformer with a custom Adaptive Med-CNN model through cross-attention fusion mechanisms for automated fetal brain MRI plane classification.

However, the authors have not fully explained how the fetal MRI images were obtained and selected. If the images consisted of those from patients with confirmed neurological anomalies or not. If the model was trained on normal fetal MRI images as well as those with confirmed anomalies. If so, what type of anomalies were included or excluded as part of the training process and selection criteria. Also, how well did the model perform in confirming or excluding images with/without anomalies present.

Reviewer #2: Summary

This study presents FetCAT, a hybrid architecture combining a Swin Transformer and a custom AdaptiveMed-CNN via cross-attention for automated classification of fetal brain MRI planes (axial, coronal, sagittal). Using 52,561 motion-degraded fetal MRI slices from 19–39 weeks’ gestation, the authors compare FetCAT with various CNN and transformer models, analyze the effect of data augmentation, and employ Grad-CAM for explainability. The proposed Swin–AdaptiveMedCNN achieved 98.64% accuracy without augmentation, outperforming all other tested models.

Major Comments

1.Generalizability and Validation

The dataset comes entirely from a single institution (Stanford LPCH), which introduces potential bias related to imaging protocol and demographics. The authors should test the model on an external dataset or, if not possible, define a fully independent test cohort held out from the start. Clearly describe how subjects were split to prevent overlapping slices between training and validation.

2.Statistical Analysis

The manuscript reports very high accuracies but does not include uncertainty estimates or statistical comparisons. Confidence intervals, per-class metrics, and statistical tests (e.g., McNemar or bootstrap confidence intervals) are needed to support statements that the proposed method significantly outperforms baselines.

3.Interpretation of Data Augmentation Results

The conclusion that augmentation reduces performance is unusual. The authors should present per-augmentation results, quantify performance differences, and discuss whether certain transformations (e.g., small-angle rotations) might still provide benefits for subsets of data such as early gestational ages or specific motion levels.

4.Data and Code Availability

The paper lists data as available in the manuscript, yet it relies on a repository that may have access restrictions. Please ensure compliance with PLOS ONE’s open-data policy by providing a link to the dataset or a clear process for accessing it. Include source code and train/validation split definitions in a public repository.

5.Ethics Statement

Although the dataset is anonymized, fetal MRI is human-subject data and typically requires an institutional review board statement or waiver. The authors should clarify whether the dataset was covered under an approved secondary-use protocol.

6.Clinical Framing

The Grad-CAM explanation is helpful and well validated by expert review. It would strengthen the paper to report clinical or workflow relevance—for example, whether the model reduces labeling time or shows consistent behavior across gestational age ranges and cases with significant motion.

7.Reproducibility Details

Include full training details such as batch size, number of epochs, early stopping criteria, and exact k-fold parameters. Provide layer sizes, dropout rates, and normalization methods. This will help others reproduce the results.

Minor Comments

•Provide separate accuracy, precision, recall, and F1 for each plane (axial, coronal, sagittal) and include a confusion matrix.

•Include calibration plots or reliability scores if the model outputs probabilities.

•Correct typographical and formatting issues (for example, “architechture” → “architecture,” “Saggital” → “Sagittal”).

•Ensure consistent use of terms such as “cross-attention” and capitalization of plane names.

•Clarify class distribution in the training and validation sets.

Recommendation

Major Revision.

The approach is technically sound and well-motivated, but the manuscript requires stronger validation, additional statistical analysis, and clarification of ethics and data availability before it can be recommended for publication.

Reviewer #3: Notes:

1- In abstract section some sentences are very long and complex, packing multiple ideas together.

2- The introduction section is generally well-structured and effectively builds a research case. But the problem statements are long and complex, shorter, focused sentences could improve readability. The novelty is mentioned late in the section, weakening the early impact and lacks a concise research question or hypothesis.

3- The literature review is extensive; the authors should provide more critical synthesis of previous work, highlighting how FetCAT directly addresses the identified gaps.

4- In methodology section many equations and algorithmic steps could be summarized conceptually.

5- The methodology section page 10 and page 9, there are opposite meaning in equation (1).

6- In methodology section, what was the specific 'k' in k-fold cross validation? What was the batch size? These are important for reproducibility.

7- In Table 5: The table is untidy. The "Swin ConvNext" row is missing "Without Augmentation" values, which looks like an error. The "AdaptiveMed" row without a transformer prefix is confusing, is this the standalone AdaptiveMed-CNN? If so, it should be in Table 3.

8- The results are presented as point estimates like ( 98.64% accuracy) without any measures of statistical significance or variance.

9- The discussion section like a summary of results rather than a realy discussion. It reports what was found but does not synthesize these findings into a higher-level argument or model for why FetCAT works so well.

10- There is no clear separation between “Discussion” and “Conclusion” both are merged into a single continuous text, which blurs their respective purposes.

6. PLOS authors have the option to publish the peer review history of their article (what does this mean?). If published, this will include your full peer review and any attached files.

Reviewer #1: No

Reviewer #2: No

Reviewer #3: No

You may also use PLOS’s free figure tool, NAAS, to help you prepare publication quality figures: https://journals.plos.org/plosone/s/figures#loc-tools-for-figure-preparation

---

## [Author Response · Author response to Decision Letter 1]

22 Nov 2025

1. Concern: Address all the remarks raised by the three reviewers

Response: We sincerely thank the reviewers and the editor for their constructive feedback. We have thoroughly addressed all comments and suggestions from all three reviewers and incorporated the required revisions throughout the manuscript.

2. Concern: Specify what kind of data you employed in your trials

Response: We appreciate the clarification request. The trials employed an open-source fetal brain MRI dataset for plane classification. Comprehensive details regarding the acquisition source, sample size, class distributions, and preprocessing steps have now been clearly provided in Section 3.1 of the methodology.

3. Concern: Highlight the steps of the proposed methods that are "explainable"

Response: Section 3.5 has been fully revised to explicitly highlight the explainable components of the proposed method. We have clarified the post-hoc Grad-CAM workflow and detailed the specific stages of the model that contribute to interpretability.

Reviewer 1:

The authors have presented a well-written manuscript highlighting a novel hybrid architecture that integrates a pre-trained Swin Transformer with a custom Adaptive Med-CNN model through cross-attention fusion mechanisms for automated fetal brain MRI plane classification. However, the authors have not fully explained how the fetal MRI images were obtained and selected. If the images consisted of those from patients with confirmed neurological anomalies or not. If the model was trained on normal fetal MRI images as well as those with confirmed anomalies. If so, what type of anomalies were included or excluded as part of the training process and selection criteria. Also, how well did the model perform in confirming or excluding images with/without anomalies present.

Response: Thank you for your insightful feedback on dataset transparency, which we agree is essential for evaluating clinical generalizability in fetal neuroimaging.

We have addressed and highlighted this by adding a paragraph to Section 3.1 ("Data Collection and Preprocessing"), clarifying that the open source dataset used in this study includes only developmentally normal fetal MRIs from 741 routine examinations (no anomalies). Since the dataset comprises exclusively developmentally normal fetal brain MRIs with no anomalous cases included for model evaluation. However, in the future, we plan to collect data specifically focusing on different anomalies in mind to enhance the generalizability of our study.

Reviewer 2.

1.

Generalizability and Validation

The dataset comes entirely from a single institution (Stanford LPCH), which introduces potential bias related to imaging protocol and demographics. The authors should test the model on an external dataset or, if not possible, define a fully independent test cohort held out from the start.

Clearly describe how subjects were split to prevent overlapping slices between training and validation.

Response: We thank the reviewer for this insightful concern. We worked on this concern carefully. To address this concern, we tested FetCAT proposed model on an entirely external dataset from OpenNeuro Fetal MRI repository (https://openneuro.org/datasets/ds003090/versions/1.0.0/metadata ), where also our model maintained high performance (81.0% accuracy), confirming its generalizability beyond the single-institution Stanford data. We have added the results and statistical analysis on this in Section 4.1.3 on our revised manuscript.

We have clearly documented that the analysis uses a subject-level split across the 741 unique subjects for 2-fold cross-validation, explicitly guaranteeing that no patient's slices overlap between the training and validation sets to prevent data leakage in updates Section 3.1.1.

2.

Statistical Analysis

The manuscript reports very high accuracies but does not include uncertainty estimates or statistical comparisons. Confidence intervals, per-class metrics, and statistical tests (e.g., McNemar or bootstrap confidence intervals) are needed to support statements that the proposed method significantly outperforms baselines.

Response: Thank you for this valuable suggestion regarding statistical validation. In response, we have now incorporated comprehensive statistical analyses including 95% confidence intervals, per-class performance metrics, and McNemar's test to rigorously validate performance differences. These additions are detailed in Section 4.1.2, and 4.1.3 with supporting result Tables. These analysis provide robust statistical evidence supporting our performance claims and enhance the reliability of our findings.

3

Interpretation of Data Augmentation Results:

The conclusion that augmentation reduces performance is unusual. The authors should present per-augmentation results, quantify performance differences, and discuss whether certain transformations (e.g., small-angle rotations) might still provide benefits for subsets of data such as early gestational ages or specific motion levels.

Response: We thank the reviewer for this valuable comment. In the revised manuscript (Section 4.3), we have added group-wise augmentation results, covering geometric, intensity-based, and noise/deformation based augmentation ablation test and quantified their performance differences to clarify the observed trends. However, as part of our future extensions, we plan to investigate augmentation effects in greater detail across gestational ages and varying motion levels applying other strategies too.

4

Data and Code Availability:

The paper lists data as available in the manuscript, yet it relies on a repository that may have access restrictions. Please ensure compliance with PLOS ONE’s open-data policy by providing a link to the dataset or a clear process for accessing it. Include source code and train/validation split definitions in a public repository.

Response: We thank the reviewer for highlighting the importance of open data and code compliance. In response, we have ensured full adherence to PLOS ONE's policy by providing direct, unrestricted access to the fetal MRI dataset through the Stanford Digital Repository, where all 52,561 anonymized images are immediately available for download. Additionally, we have made our complete implementation publicly available in the FetCAT repository, which includes the hybrid Swin Transformer-CNN architecture, training pipeline, and validation split definitions, thus ensuring full transparency and reproducibility of our study.

5

Ethics Statement

Although the dataset is anonymized, fetal MRI is human-subject data and typically requires an institutional review board statement or waiver. The authors should clarify whether the dataset was covered under an approved secondary-use protocol.

Response: Thank you for raising this important point on ethics for human-subjects data, which underscores the need for clear IRB transparency.

We have added a new subsection at the end detailing that the dataset was collected under Stanford University's IRB protocol with informed consent, and its public, anonymized release permits secondary use without additional approval at our institution.

6

Clinical Framing

The Grad-CAM explanation is helpful and well validated by expert review. It would strengthen the paper to report clinical or workflow relevance—for example, whether the model reduces labeling time or shows consistent behavior across gestational age ranges and cases with significant motion

Response: We thank the reviewer for this valuable suggestion. In response, we have now added a new Figure 6 that illustrates the clinical workflow integration, and a corresponding discussion in Section 4.2.1, explicitly addressing the model's reduction in labeling time and its consistent performance across gestational ages and motion-degraded cases.

7

Reproducibility Details

Include full training details such as batch size, number of epochs, early stopping criteria, and exact k-fold parameters. Provide layer sizes, dropout rates, and normalization methods. This will help others reproduce the results.

Response: Thank you for this valuable suggestion. We have added a comprehensive subsection in the methodology detailing all training hyperparameters, architectural specifications, and preprocessing steps to ensure full reproducibility of our results. Additionally, the complete implementation code has been made publicly available and mentioned at the end of the manuscript to facilitate replication of our experiments.

Minor Comments

8

•Provide separate accuracy, precision, recall, and F1 for each plane (axial, coronal, sagittal) and include a confusion matrix.

Thank you for the insightful concern. To address the reviewer's feedback, we added a new Table in the Results section with class-wise metrics (accuracy, precision, recall, F1-score) for Axial, Coronal, and Sagittal planes, including point estimates and 95% CIs from cross-validation. We also included a Figure showing the 2-fold CV and test set confusion matrices, which demonstrate strong performance with few misclassifications with the proposed technique.

9

•Include calibration plots or reliability scores if the model outputs probabilities.

We addressed the reviewer's concern by calculating and including the model's calibration plots and scores with visualization in Section 4.1.2, which confirmed the reliability of the predicted probabilities.

10

•Correct typographical and formatting issues (for example, “architechture” → “architecture,” “Saggital” → “Sagittal”).

We thank the reviewer for careful reading; all noted typographical errors, such as "architechture" and "Saggital," have been corrected throughout the manuscript.

11

•Ensure consistent use of terms such as “cross-attention” and capitalization of plane names.

We have ensured consistent hyphenation and lowercase for 'cross-attention' throughout the manuscript, standardized capitalization of plane names (e.g., 'Axial', 'Coronal', 'Sagittal') as class labels

12

•Clarify class distribution in the training and validation sets.

We thank the reviewer for the suggestion and have acknowledged the class distribution in the revised manuscript in section 3.1.1.

Reviewer 3’s Comments to the Author:

Reviewer’s Concerns

Author’s Responses

1

In abstract section some sentences are very long and complex, packing multiple ideas together.

Response: Thank you for your observation. We have revised the abstract to break down complex sentences into simpler, more focused statements, ensuring each sentence conveys a single clear idea for improved readability.

2

The introduction section is generally well-structured and effectively builds a research case. But the problem statements are long and complex, shorter, focused sentences could improve readability. The novelty is mentioned late in the section, weakening the early impact and lacks a concise research question or hypothesis.

Response: We thank the reviewer for the constructive feedback. The introduction has been revised with shorter, more focused sentences, an earlier emphasis on our novel FetCAT architecture, and the explicit inclusion of our central hypothesis on cross-attention fusion.

3

The literature review is extensive; the authors should provide more critical synthesis of previous work, highlighting how FetCAT directly addresses the identified gaps.

Response: We thank the reviewer for this valuable suggestion. We agree that a stronger critical synthesis strengthens the narrative for our proposed method. In direct response to this comment, we have thoroughly revised the final two paragraphs of the Literature Review (Section 2).

4

In methodology section many equations and algorithmic steps could be summarized conceptually.

Response: We appreciate the reviewer for this suggestion. We have added a conceptual summary for clarity but retain the algorithms to ensure the reproducibility and precise implementation of our novel cross-attention fusion mechanism, which is a core contribution of this work.

5

The methodology section page 10 and page 9, there are opposite meaning in equation (1).

Response: Thank you for your careful observation regarding the inconsistency in Equation (1) between pages 9 and 10 of the methodology section; it was a mistake and we have revised it to align the meanings consistently across both cases.

6

In methodology section, what was the specific 'k' in k-fold cross validation? What was the batch size? These are` important for reproducibility.

Response: Thank you for this valuable suggestion. We have added a comprehensive subsection in the methodology detailing all training hyperparameters, architectural specifications, and preprocessing steps to ensure full reproducibility of our results. Additionally, the complete implementation code has been made publicly available and mentioned at the end of the manuscript to facilitate reproducibility of our experiments.

7

In Table 5: The table is untidy. The "Swin ConvNext" row is missing "Without Augmentation" values, which looks like an error. The "AdaptiveMed" row without a transformer prefix is confusing, is this the standalone AdaptiveMed-CNN? If so, it should be in Table 3.

Response: Thank you for highlighting this formatting issue in the table; the "AdaptiveMed" row was intended as the CNN backbone shared across hybrid fusions (e.g., BEiT-AdaptiveMed and DeiT-AdaptiveMed) to enable direct comparisons of transformer variants on a consistent local feature extractor, but line wrapping caused it to appear standalone. The omission of "Without Augmentation" values for the "Swin-ConvNeXt" row was an unintentional error during table preparation, which we have now corrected by adding the computed metrics. We have reformatted the table for clarity.

8

The results are presented as point estimates like ( 98.64% accuracy) without any measures of statistical significance or variance.

Response: Thank you for this valuable suggestion regarding statistical validation. In response, we have now incorporated comprehensive statistical analyses including 95% confidence intervals, per-class performance metrics, and McNemar's test to rigorously validate performance differences. These additions are detailed in Section 4.1.2, and 4.1.3 with supporting result Tables. These analysis provide robust statistical evidence supporting our performance claims and enhance the reliability of our findings.

9

The discussion section like a summary of results rather than a realy discussion. It reports what was found but does not synthesize these findings into a higher-level argument or model for why FetCAT works so well.

Response: We have addressed this concern by restructuring the manuscript and creating a separate discussion section that synthesizes the findings and provides a higher-level interpretation of why FetCAT performs effectively.

10

There is no clear separation between “Discussion” and “Conclusion” both are merged into a single continuous text, which blurs their respective purposes.

Response: We acknowledge this fact and addressed the reviewer's feedback by creating two distinct and focused sections: "Discussion" and "Conclusion", ensuring clear separation of results synthesis from the final summary statements in our revised manuscript.

---

## [Decision Letter · Decision Letter 1]

18 Dec 2025

FetCAT: Cross-Attention Fusion of Transformer-CNN Architecture for Fetal Brain Plane Classification with Explainability using Motion-degraded MRI

PONE-D-25-47910R1

Dear Dr. Suha,

We’re pleased to inform you that your manuscript has been judged scientifically suitable for publication and will be formally accepted for publication once it meets all outstanding technical requirements.

Kind regards,

Alessandro Bruno, Ph.D.

Academic Editor

PLOS One

Additional Editor Comments (optional):

Dear Authors,

I am glad to let you know that I appreciate your efforts to improve the manuscript's quality.

I will, therefore, recommend it for acceptance.

With regards,

A.B.

Reviewers' comments:

Reviewer's Responses to Questions

**Comments to the Author**

1. If the authors have adequately addressed your comments raised in a previous round of review and you feel that this manuscript is now acceptable for publication, you may indicate that here to bypass the “Comments to the Author” section, enter your conflict of interest statement in the “Confidential to Editor” section, and submit your "Accept" recommendation.

Reviewer #1: All comments have been addressed

2. Is the manuscript technically sound, and do the data support the conclusions?

Reviewer #1: (No Response)

3. Has the statistical analysis been performed appropriately and rigorously?

Reviewer #1: (No Response)

4. Have the authors made all data underlying the findings in their manuscript fully available?

Reviewer #1: (No Response)

5. Is the manuscript presented in an intelligible fashion and written in standard English?

Reviewer #1: (No Response)

6. Review Comments to the Author

Reviewer #1: (No Response)

7. PLOS authors have the option to publish the peer review history of their article (what does this mean?). If published, this will include your full peer review and any attached files.

Reviewer #1: No

---

## [Editor Report · Acceptance letter]

PONE-D-25-47910R1

PLOS One

Dear Dr. Suha,

I'm pleased to inform you that your manuscript has been deemed suitable for publication in PLOS One. Congratulations! Your manuscript is now being handed over to our production team.

Kind regards,

on behalf of

Associate Professor Alessandro Bruno

Academic Editor

PLOS One